# The dark kinase STK32A regulates hair cell planar polarity opposite of EMX2 in the developing mouse inner ear

Shihai Jia[1†], Evan M Ratzan[2,3†], Ellison J Goodrich[1], Raisa Abrar[1], Luke Heiland[4], Basile Tarchini[5,6], Michael R Deans[1,4*]

[1]Department of Neurobiology, Spencer Fox Eccles School of Medicine at the University of Utah, Salt Lake City, United States; [2]Interdepartmental Program in Neuroscience, Spencer Fox Eccles School of Medicine at the University of Utah, Salt Lake City, United States; [3]Departments of Otolaryngology and Neurology, Boston Children's Hospital and Harvard Medical School, Boston, United States; [4]Department of Otolaryngology, Spencer Fox Eccles School of Medicine at the University of Utah, Salt Lake City, United States; [5]The Jackson Laboratory, Bar Harbor, United States; [6]Tufts University School of Medicine, Boston, United States

*For correspondence:
michael.deans@utah.edu

†These authors contributed equally to this work

Competing interest: The authors declare that no competing interests exist.

**Abstract** The vestibular maculae of the inner ear contain sensory receptor hair cells that detect linear acceleration and contribute to equilibrioception to coordinate posture and ambulatory movements. These hair cells are divided between two groups, separated by a line of polarity reversal (LPR), with oppositely oriented planar-polarized stereociliary bundles that detect motion in opposite directions. The transcription factor EMX2 is known to establish this planar polarized organization in mouse by regulating the distribution of the transmembrane receptor GPR156 at hair cell boundaries in one group of cells. However, the genes regulated by EMX2 in this context were previously not known. Using mouse as a model, we have identified the serine threonine kinase STK32A as a downstream effector negatively regulated by EMX2. *Stk32a* is expressed in hair cells on one side of the LPR in a pattern complementary to *Emx2* expression in hair cells on the opposite side. *Stk32a* is necessary to align the intrinsic polarity of the bundle with the core planar cell polarity (PCP) proteins in EMX2-negative regions, and is sufficient to reorient bundles when ectopically expressed in neighboring EMX2-positive regions. We demonstrate that STK32A reinforces LPR formation by regulating the apical localization of GPR156. These observations support a model in which bundle orientation is determined through separate mechanisms in hair cells on opposite sides of the maculae, with EMX2-mediated repression of *Stk32a* determining the final position of the LPR.

## Editor's evaluation

This important study provides a significant advance in the understanding of the molecular mechanisms that establish planar cell polarity in hair cells of the mammalian inner ear. The conclusions, which are supported by compelling evidence, will be of interest to those studying the development and function of the vestibular system, and planar cell polarity.

## Introduction

Planar polarity is an organizational feature of epithelia in which the polarity of cells and tissues is coordinated along a plane perpendicular to the apical-basal cell polarity axis. Planar polarity is seen across diverse species and tissues, and this organization is essential for early developmental events

including left/right asymmetry in the primitive streak and coordinated cellular movements such as convergent extension and neural tube closure (*Minegishi et al., 2017*; *Roszko et al., 2009*; *Williams and Solnica-Krezel, 2020*). Similarly, within the inner ear the function of the sensory receptor hair cells that detect sound and motion is dependent upon the development of a planar polarized bundle of elongated microvilli called stereocilia that project from the apical cell surface (*Deans, 2021*; *Tarchini and Lu, 2019*). Mutant mice with disrupted planar polarity have vestibular deficits indicating that planar polarity is required for hair cell function (*Duncan et al., 2017*; *Simon et al., 2021*).

The hair cell stereociliary bundle is comprised of rows of stereocilia of increasing length with the tallest row positioned alongside the kinocilium, a true tubulin-based cilium located at one side of the hair cell apical surface (*Schwander et al., 2010*). When the bundle is deflected by overlying extracellular matrices, mechanically gated ion channels are opened by tip-links connecting the ion channels on the tops of shorter stereocilia to the shafts of their taller neighbors. Due to the organization of tip-links and ion channels, only deflections of the bundle towards the kinocilium are excitatory (*Gillespie and Müller, 2009*; *Hudspeth, 1989*). As a result, hair cell planar polarity and function are linked. Planar polarity is further specialized in the utricle and saccule; the two vestibular maculae that detect linear acceleration. Hair cells in these maculae are divided between two groups with oppositely oriented stereociliary bundles and as a result, detect linear acceleration occurring in opposite directions. In the utricle hair cells are oriented with their kinocilia positioned towards each other while in the saccule they are positioned away, and the cell boundary separating these groups is called the Line of Polarity Reversal (LPR) (*Deans, 2013*). The afferent innervation of the maculae is also coordinated with the LPR and afferent neurons selectively contact hair cells located on either one side of the LPR or the other (*Ji et al., 2022*; *Maklad et al., 2010*). A similar organization of hair cells and neurons underlies the detection of fluid flow in lateral line neuromasts of teleost fish where it has been referred to as planar bipolarity (*Kozak et al., 2020*; *Lozano-Ortega et al., 2018*).

Planar polarity and formation of the LPR in the vestibular maculae is dependent upon three molecular events. The first is the intrinsic polarization of the hair cell which leads to the lateral placement of the kinocilium and formation of the staircase array of stereocilia. Intrinsic polarity is mediated in part by the small GTPase Rac1 and its effector Pak, which stabilize interactions between apical microtubules, the basal body and adjacent hair cell junction (*Grimsley-Myers et al., 2009*; *Sipe et al., 2013*), and the G-protein signaling modulators GPSM2 and GNAI which shape the stereociliary bundle and specify the tallest stereocilia (*Tadenev et al., 2019*; *Tarchini et al., 2016*). The second is planar cell polarity (PCP) signaling between neighboring cells which coordinates the orientation of their stereociliary bundles and orients them along a shared axis within the sensory epithelium. PCP is established and maintained by the core PCP proteins, which are asymmetrically distributed within individual cells and mediate the exchange of polarity signals between neighboring hair cells (*Stoller et al., 2018*). As a result, in PCP mutants, intrinsic hair cell polarity remains intact while the bundles are misoriented relative to each other and within the sensory epithelia (*Duncan et al., 2017*; *Curtin et al., 2003*; *Montcouquiol et al., 2003*; *Wang et al., 2006*; *Yin et al., 2012*). The asymmetric distribution of the core PCP proteins remains constant across the LPR as revealed by the relative distributions of the PCP proteins Prickle-like 2 (PK2) and Frizzled 3 (FZD3) (*Deans et al., 2007*). The third event is mediated by the regional expression of the transcription factor EMX2 which creates the LPR by reversing the orientation of one group of hair cells relative to the PCP axis (*Jiang et al., 2017*). As a result, Emx2-expressing hair cells have the kinocilium positioned adjacent to PK2 while hair cells without EMX2 position the kinocilium opposite of PK2. EMX2 works in conjunction with the G-protein coupled receptor GPR156 to establish this bundle orientation in the lateral region of the utricle and central region of the saccule (*Kindt et al., 2021*). In both *Emx2* and *Gpr156* mutants, intrinsic polarization of the bundle remains intact but these bundles are uniformly oriented and fail to form an LPR. However, despite this functional relationship, GPR156 is not transcriptionally regulated by EMX2. Instead GPR156 function is regulated post-translationally, and is only present at the apical cell boundaries in hair cells expressing EMX2 where it acts to reorient the intrinsic polarity relative to the PCP axis (*Kindt et al., 2021*).

Although EMX2 is a transcription factor that acts like a molecular switch to pattern hair cells and establish the LPR, the transcriptional targets acting downstream of EMX2 have not been identified. Moreover, additional molecules that might position the kinocilium relative to the PCP axis in EMX2-negative hair cells are not known. We addressed these knowledge gaps using transcriptional profiling

of *Emx2* mutant utricles and identified the serine-threonine kinase gene *Stk32a/Yank1* which is expressed in the medial utricle and is actively repressed in lateral regions expressing EMX2. STK32A is a dark kinase, an enzyme with previously unknown biological function or substrates (*Berginski et al., 2021*). Here we provide genetic evidence that STK32A regulates planar polarity, and that in vestibular hair cells STK32A aligns the intrinsic polarity of the stereociliary bundle with the intercellular PCP axis in hair cells that do not express EMX2. In addition, we provide evidence that STK32A regulates the subcellular distribution of GPR156 which prevents this receptor from functioning in the absence of EMX2, thereby establishing the position of the LPR.

## Results
### Stk32a is expressed in the absence of EMX2

The planar polarity of vestibular hair cells in the utricle and saccule of the mouse inner ear is established by the overlapping function of the core PCP proteins which coordinate stereociliary bundle orientation along a shared polarity axis, and the transcription factor EMX2 which determines the orientation of the stereociliary bundle relative to that PCP polarity axis (*Deans, 2021*; *Jiang et al., 2017*). EMX2 expression is restricted to the lateral region of the utricular sensory epithelium and the inner region of the saccular sensory epithelium, and the hair cells in these regions orient their bundles such that the kinocilium is positioned adjacent to the PCP protein PK2, and opposite of Frizzled. In contrast, the kinocilium in EMX2-negative hair cells of the other regions are positioned opposite of PK2 and therefore adjacent to Frizzled (*Deans, 2021*; *Deans et al., 2007*; *Jiang et al., 2017*). As a result, the vestibular maculae contain two groups of hair cells with opposite bundle orientations that meet at the LPR (*Figure 1A–D*). EMX2 directly regulates bundle orientation relative to the underlying PCP axis because all utricular hair cells adopt the orientation of medial hair cells in *Emx2* mutant mice and the orientation of lateral hair cells following EMX2 overexpression (*Jiang et al., 2017*). Genes regulated by EMX2 during vestibular development in the mouse were identified using a bulk RNAseq approach in which the transcriptional profile of *Emx2* mutant utricles was compared to utricles dissected from wild type littermate controls (*Figure 1H*). Mutants were derived from the *Emx2*-Cre knock-in line in which the *Emx2* coding sequence is replaced by Cre recombinase (*Kimura et al., 2005*). In these *Emx2* [Cre/Cre] mice, EMX2 function is lost and all utricular hair cells are oriented in a common direction (*Figure 1E–G*) similar to that originally reported in *Emx2* knockout (KO) mice (*Holley et al., 2010*).

RNAseq revealed significant changes in gene expression in *Emx2* [Cre/Cre] utricles that included the downregulation of genes that are presumed to be direct or indirect transcriptional targets of EMX2 (*Supplementary file 1*). Despite this, the majority of genes with statistically significant changes in expression levels were still present and did not have changes in their spatial distribution in *Emx2* [Cre/Cre] utricles when compared to controls by in situ hybridization techniques. Nonetheless, several genes that were downregulated or missing included *Slc26a4*, *Caln1* and *Stk26/Mst1* (*Figure 1—figure supplement 1* 1). These genes are found in the transitional epithelium, an EMX2-expressing region adjacent to the sensory epithelia that does not contain hair cells but may contain hair cell progenitors (*Burns et al., 2015*; *Jan et al., 2021*) and contributes to maintaining the ionic composition of endolymph (*Kim and Marcus, 2011*). Since these genes are not found in utricular hair cells, they are not likely to encode regulators of bundle orientation.

Genes that show increased expression in *Emx2* [Cre/Cre] utricles are likely to be transcriptionally repressed by EMX2, through either direct or indirect mechanisms, and may also contribute to hair cell patterning about the LPR (*Supplementary file 2*). One such candidate was *Stk32a/Yank1* which is expressed in the medial region of the utricle and outer region of the saccule in a pattern complementary to that of *Emx2* when visualized by fluorescent in situ hybridization (fISH) (*Figure 1I and J*). STK32A is a serine threonine kinase with sequence similarity to AGC kinases and is a member of a small kinase protein family that includes STK32B/YANK2 and STK32C/YANK3. The STK32 kinases have been categorized as dark kinases because their substrates and biological functions are not known (*Berginski et al., 2021*). Within the developing utricle, *Stk32a* mRNA was detected in clusters that overlapped with mRNA transcribed from the hair cell gene *Gfi1* (*Figure 1K*). A similar pattern of expression was seen in the semi-circular canal cristae (*Figure 1L and M*). The hair cells in each of these regions lack EMX2 and have stereociliary bundles oriented with the kinocilium positioned opposite of PK2 similar to utricular hair cells located medial to the LPR (*Deans et al., 2007*).

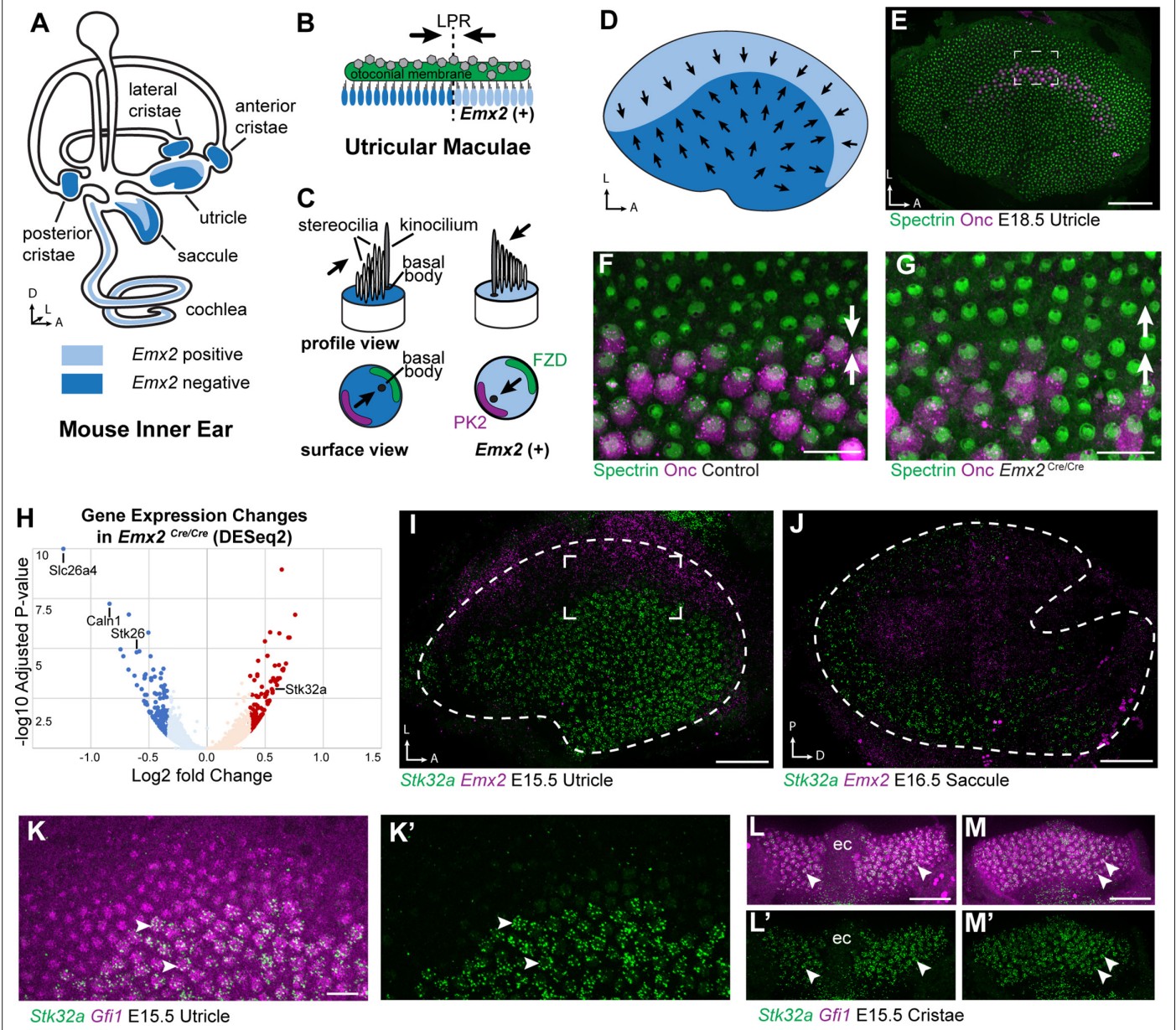

**Figure 1.** Transcriptome analysis of *Emx2* mutant utricles identifies *Stk32a* as a candidate planar polarity effector. (**A**) The sensory epithelia (blue shading) of the mouse inner ear contains hair cells that can be divided into those that express EMX2 and those that do not. (**B**) The utricle contains two groups of hair cells patterned about the LPR and oriented to detect movements of the otoconial membrane directed in opposite directions (arrows). (**C**) Utricular hair cells with opposite stereociliary bundle orientations (arrows) viewed in profile or from a top-down surface view. The cuticular plate (blue) can be used to determine bundle orientation based upon the position of the basal body. The Core PCP proteins PK2 and FZD are similarly distributed in hair cells with either bundle orientation. (**D**) The border of EMX2 expression distinguishes hair cells with opposite bundle orientations (arrows). (**E**) E18.5 mouse utricle labeled for βII-Spectrin to mark the hair cell cuticular plates and Oncomodulin to mark the striolar region. (**F**) Higher magnification of the framed region from 'E' illustrating the orientation (arrows) of the two hair cell groups. (**G**) In *Emx2* ^Cre/Cre^ mutants the hair cells are oriented in a single direction. (**H**) Gene expression changes in *Emx2* ^Cre/Cre^ utricles relative to littermate control determined by bulk RNAseq analysis of micro-dissected tissue. Select upregulated and downregulated genes evaluated in subsequent figures are annotated. (**I**) Fluorescent ISH showing the complementary patterns of *Emx2* and *Stk32a* mRNA expression in E15.5 utricle or (**J**) E16.5 saccule. Dashed lines mark the boundary of the sensory epithelia determined by *Gfi1* expression in a third channel. (**K,K'**) Higher magnification of the framed region in 'I' showing the cellular distribution of *Stk32a* mRNA and overlap with *Gfi1* in medial hair cells. (**L,L'**) *Stk32a* similarly colocalizes with *Gfi1* in the anterior cristae and (**M,M'**) horizontal cristae. Arrowheads highlight individual hair cells. (ec) Eminentia Cruciatum. Scale bars: E,I,J,L,M (50 μm); F,G,K (20 μm).

The online version of this article includes the following figure supplement(s) for figure 1:

**Figure supplement 1.** Down-regulated genes are enriched in TE.

The increase in *Stk32a* mRNA detected by RNAseq is due to an expansion of the *Stk32a* expression domain and in the absence of *Emx2*, *Stk32a* mRNA can be detected throughout the entire sensory epithelia (*Figure 2A and B*). This expansion was quantified from multiple biological replicates by measuring the ratio of average fluorescence intensities of fISH signal for *Stk32a* mRNA in medial versus lateral regions. When expressed as a ratio, medial *Stk32a* expression is more than 3-fold higher in littermate controls than in *Emx2* ^Cre/Cre^ utricles where this ratio becomes equally proportioned due to *Stk32a* expansion (*Figure 2D, E and H*). Conversely, when *Emx2* is ectopically expressed in hair cells throughout the sensory epithelia using hair cell-restricted *Gfi1*-Cre crossed with Lox-Stop-Lox (LSL) *Emx2* overexpression mice (*Rosa26* ^LSL-Emx2^), *Stk32a* expression is reduced wherever *Emx2* is induced (*Jiang et al., 2017*). As a result, the expression of both genes becomes balanced between the medial and lateral utricle with fluorescent expression ratios that are equally proportioned (*Figure 2F, G, I, J*). Together these gene expression studies identify *Stk32a* as a gene negatively regulated in the lateral region of the utricle resulting in non-overlapping and complementary patterns of *Emx2* and *Stk32a*.

## Stk32a overexpression is sufficient to dictate the orientation of lateral hair cells

As *Stk32a* is negatively regulated by EMX2, STK32A function in planar polarity was evaluated using an adeno-associated virus (AAV) vector to overexpress *Stk32a* mRNA in lateral hair cells of the utricle where *Stk32a* expression is normally repressed. For these experiments, utricles were micro-dissected and grown as explants prior to AAV transduction. Since the AAV vector does not contain the genomic regulatory elements surrounding *Stk32a* it is not subject to EMX2 associated repression. Transduced cells were identified based upon EGFP expression from a bicstronic *Stk32a-P2A-EGFP* mRNA (*Figure 3B–E*). Following 48 hr in culture post-transduction, bundle orientation was determined using βII-Spectrin immunolabeling of the cuticular plate to visualize the fonticulus and thereby infer the position of the kinocilium and orientation of the bundle. Since culture conditions often distorted explant growth, coarse measures of bundle orientations were used that defined medial and lateral orientations based upon position of the fonticulus within quadrants on the medial or lateral apical cell surface (*Figure 3F*). Hair cells with a fonticulus located in one of the two remaining quadrants of the cell surface were grouped as misoriented. βII-SPECTRIN labeling in some cells did not provide a defined fonticulus that could be used to evaluate orientation and were grouped as undetermined. Control explants were transduced with AAV expressing EGFP alone and the majority of transduced cells had bundle orientations appropriate for their position in the medial or lateral utricle (*Figure 3A,A',F*).

In contrast to controls, hair cells located in the lateral utricle transduced at P0 with AAVs expressing *Stk32a-P2A-EGFP* showed prominent changes in bundle orientation with many lateral hair cells having orientations that resembled hair cells from the medial utricle (*Figure 3B,B',F*). A smaller proportion were also misoriented relative to non-transduced neighbors though it could not be determined whether this was their final orientation or whether they were undergoing an STK32A-mediated rotation at the time of tissue fixation. When lateral hair cells were transduced at P5 with *Stk32a-P2A-EGFP* (*Figure 3E,E',F*), fewer reversed cells were present suggesting that the impact of ectopic STK32A might be age dependent. Most likely this is because STK32A acts during stereociliary bundle morphogenesis and is less capable of reorienting the bundle once the structure has matured. This effect is also likely dependent upon the kinase activity of STK32A because it is disrupted by substituting the Lysine residue that facilitates ATP-binding prior to substrate phosphorylation with Arginine (K52R) (*Sorrell et al., 2020*; *UniProt Consortium, 2021*). Consistent with this, AAV expressing *Stk32* ^K52R^*-P2A-EGFP* was incapable of reorienting hair cells in the lateral region of the utricle when introduced to utricle explants at P0 (*Figure 3C,C',F*).

Currently available antibodies against STK32A lack the specificity to distinguish STK32A from the closely related STK32B and STK32C kinases and demonstrate poor labeling of fixed tissues (data not shown). Therefore, in order to determine the sub-cellular distribution of STK32A, *Stk32a::EGFP* fusion constructs were generated and transduced into developing hair cells using AAV vectors. Since EGFP-tag placement may impact protein localization, alternative viruses expressing Amino- and Carboxy-terminal EGFP were evaluated. Carboxy-terminal EGFP tag had a greater capacity to promote reversed bundle orientations in lateral hair cells (*Figure 3—figure supplement 1*). Since STK32A::EGFP appeared functional in this assay, the distribution of EGFP immunolabeling was used as a proxy for the subcellular distribution of endogenous STK32A. Using this approach,

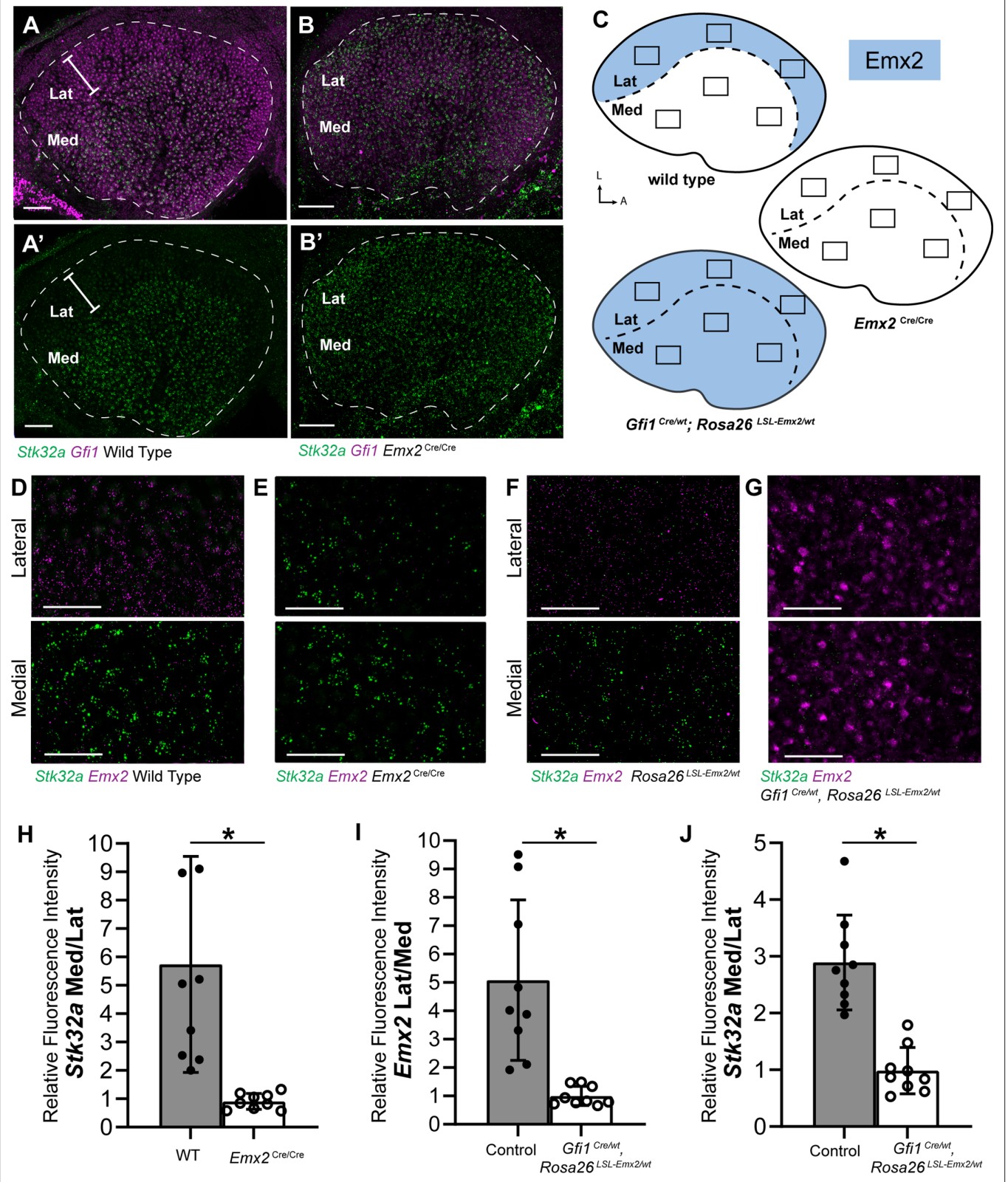

**Figure 2.** *Stk32*a expression is inversely correlated with *Emx2*. (**A,A′**) *Stk32a* mRNA is restricted to the medial region (Med) of the mouse utricle and expands in *Emx2^Cre/Cre* mutants (**B,B′**) to overlap with the hair cell marker *Gfi1* throughout the sensory epithelium. (**C**) *Emx2* distribution in wild type tissues, *Emx2* ^Cre/Cre^ mutants or following transgenic overexpression (*Gfi1* ^Cre/wt^; *ROSA* ^Emx2/wt^). Boxed regions indicate relative positions of analysis fields used for fluorescence intensity measures. (**D,E**) Representative images used to measure *Emx2* and *Stk32a* wISH fluorescence in control and *Emx2* ^Cre/Cre^

*Figure 2 continued on next page*

Figure 2 continued

mutant utricles. (**F,G**) Representative images used to measure *Emx2* and *Stk32a* wISH fluorescence from Cre-negative controls and *Gfi1*-Cre, *Rosa26* <sup>LSL-Emx2/wt</sup> transgenic mice overexpressing *Emx2* in hair cells. (**H**) Relative fluorescence for *Stk32a* in medial (Med) vs. lateral (Lat) regions following *Emx2* gene deletion. (**I**) Relative fluorescence for *Emx2* in lateral vs. medial and (**J**) *Stk32a* in medial vs. lateral regions following *Emx2* overexpression. Control mice used for quantification are Cre-negative. Pairwise comparisons of relative fluorescence intensity were evaluated by Student's test (* p<0.002) and error bars indicate SD. Scale bars: A,B (50 μm); D-G (25 μm).

STK32A::EGFP was detected in the apical compartment of transduced hair cells and stereociliary bundle (*Figure 3G, H*). STK32A::EGFP can be detected in this apical compartment between 12 hr and 24 hr post-transduction and continues to be enriched in this area despite increased STK32A::EGFP labeling throughout the hair cell over the subsequent 48 hr. This is in contrast to EGFP alone which remained cytosolic and strictly labeled the cell soma (*Figure 3A–E*). Motif-based evaluation of the STK32A primary amino acid sequence also identified the potential for myristoylation of the Glycine located at amino acid position 2. The potential for this lipid modification to localize STK32A to the apical compartment of hair cells was tested using AAV expressing a GFP fusion protein in which the N-myristoyl glycine is deleted. When evaluated in transduced hair cells STK32A^{Δ2G}::EGFP was not enriched in the stereociliary bundle (*Figure 3F,I,I',J,J'*). Moreover, STK32A^{Δ2G}::EGFP was not capable of reversing the orientation of hair cells in the lateral utricle indicating that STK32A needs to be located at the apical cell surface to have this affect (*Figure 3D, D' and F*). Together, the distribution of these STK32A::EGFP fusion proteins and the capability of STK32A::EGFP to generate hair cells with medial bundle orientations in the lateral utricle suggests that endogenous STK32A functions in the apical compartment of developing hair cells.

## STK32A coordinates intracellular polarity with the PCP axis

Since STK32A is sufficient to direct lateral hair cells of the utricle to develop reversed bundle orientations, a knockout (KO) mouse was generated using CRISPR-mediated mutagenesis to determine if *Stk32a* was also necessary for the planar polarized development of medial hair cells (*Figure 4—figure supplement 1*). This targeting strategy generated two CRISPR-mediated DNA breaks in the introns flanking *Stk32a* exon 2, which contains the translational start and N-myristoyl glycine, and deleted exon 2 following non-homologous end-joining repair. The lack of a suitable antibody against STK32A prevented Western blot analyses to determine whether a mutant protein was translated from alternative start codons. However, any mutant protein would lack the essential N-myristoyl glycine and ATP-binding motifs required for function. When the deletion was homozygosed, *Stk32a* ^{-/-} mice were viable, survived to adulthood and showed no overt physical or behavioral differences from littermate controls. Furthermore, no differences were detected in the planar polarized organization of utricular hair cells between wild type and *Stk32a* ^{+/-} littermates (*Figure 4—figure supplements 2–3*). Since this indicated that exon2 deletion did not result in a dominant-negative or hypomorphic allele, heterozygotes were used as controls for all subsequent analyses.

When evaluated by immunofluorescent labeling of βII-SPECTRIN at P5, it was evident that the planar polarized organization of vestibular hair cells in the medial regions of the *Stk32a* ^{-/-} utricle was severely disrupted (*Figure 4* and *Figure 4—figure supplement 2*). Unexpectedly, mutant hair cells in these regions did not assume the same bundle orientation of hair cells that do not normally express *Stk32a* located in the lateral region. Instead the bundles of hair cells in the striolar and medial extra-striolar regions of the *Stk32a* ^{-/-} utricle were randomly oriented relative to each other (*Figure 4E–L*) while lateral hair cells remained correctly oriented (*Figure 4A–D*). Misoriented bundles were also seen in the outer region of the saccule (*Figure 4M–P*) and within the semi-circular canal cristae (*Figure 4Q-Bb*); regions that normally express *Stk32a* and not *Emx2*. Cumulative measures of bundle orientation plotted as circular histograms confirmed this phenotype across biological replicates and demonstrated that bundle orientation was disorganized rather than reversed (*Figure 4*). The location of analysis fields used for these measures relative to domains of *Emx2* and *Stk32a* expression are summarized schematically (*Figure 4—figure supplement 2*). In general, the extent of misorientation was greater in the maculae than in the cristae though the basis for this difference is unknown. Auditory hair cells were not impacted in *Stk32a* ^{-/-} cochleae (*Figure 4—figure supplement 4*), most likely because they express *Emx2* and not *Stk32a* (*Jiang et al., 2017*; *Orvis et al., 2021*). Despite the randomization phenotype, the intrinsic polarity of the majority of vestibular hair cells remained intact

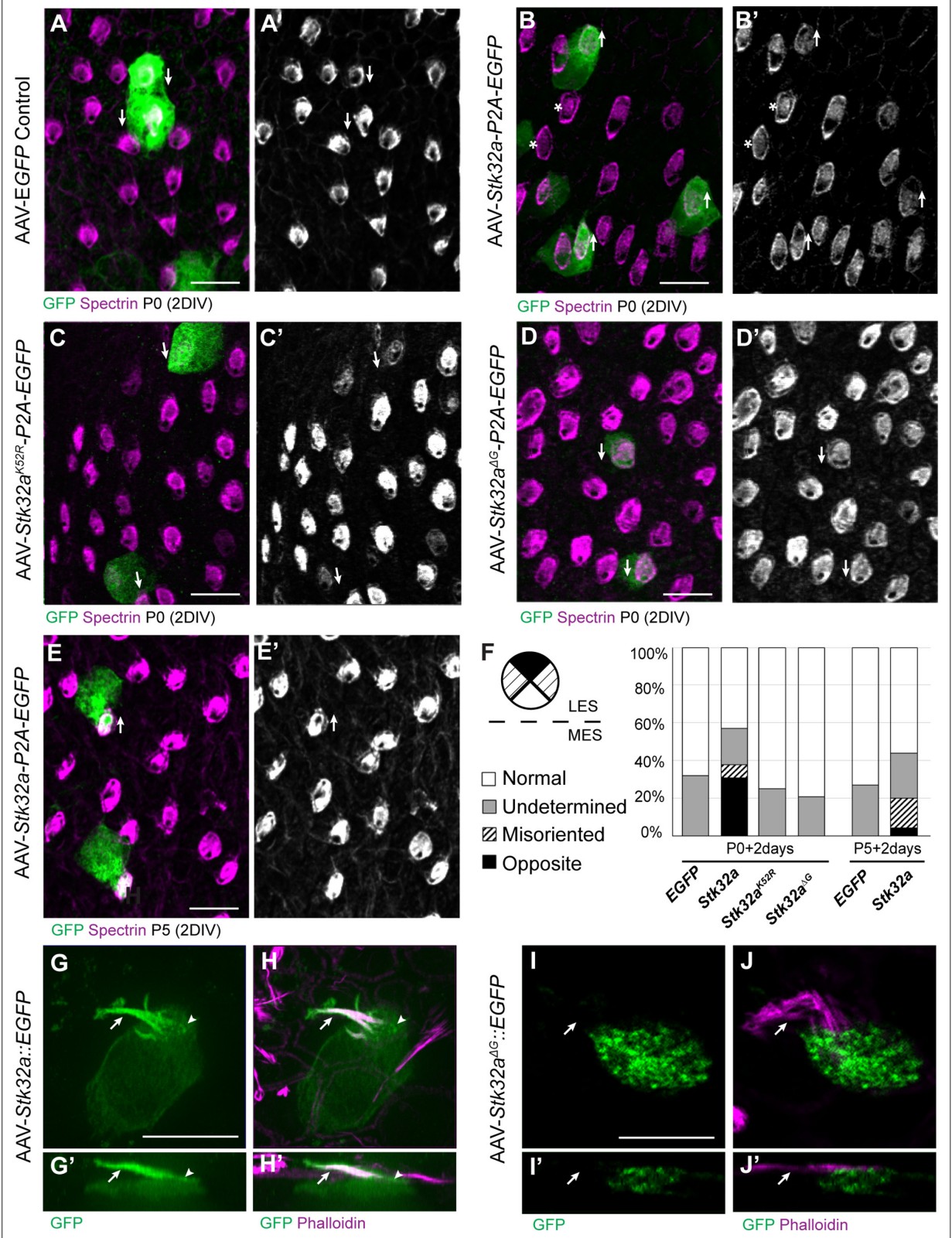

**Figure 3.** STK32A is sufficient to reorient stereociliary bundles in the Emx2 domain. (**A,A'**) Hair cells from the lateral region transduced with AAV vectors expressing EGFP at P0 develop a cuticular plate with the fonticulus positioned towards the LPR when evaluated by βII-Spectrin labeling. Arrows illustrate orientation of transduced cells. (**B,B'**) Hair cells from the lateral region expressing STK32A and EGFP frequently developed medial bundle orientations as revealed by the position of the fonticulus. Arrows illustrate orientation of transduced cells and asterisks mark examples of non-transduced cells with

*Figure 3 continued on next page*

*Figure 3 continued*

undetermined bundle orientation. (**C,C′**) Hair cells from the lateral region expressing the kinase mutant STK32A [K52R]. (**D,D′**) Hair cells from the lateral region expressing the STK32A [Δ2G] mutant lacking the predicted N-myristoyl glycine. (**E,E′**) Medial bundle orientations are less frequent in lateral hair cells transduced at P5. (**F**) Frequency of reoriented or misoriented stereociliary bundles for AAV transduced hair cells in the lateral extrastriolar of utricular explants cultured from P0 and P5 mice. Cells with a fonticulus located in the apical cell surface quadrant closest to the LPR were considered normal while cells with a fonticulus in the quadrant furthest from the LPR were classified as Opposite. Cells with a poorly resolved fonticulus were classified as Undetermined while all other orientations were considered Misoriented for this quantification. For *AAV-Stk32a-P2A-EGFP* at P0 (N=433 cells from 5 utricles), *AAV-Stk32a-P2A-EGFP* at P5 (N=109 cells from 4 utricles), *AAV-EGFP* at P0 (N=167 cells from 3 utricles), *AAV-EGFP* at P5 (N=165 cells from 3 utricles), *AAV-Stk32a [Δ2G]-P2A-EGFP* at P0 (N=106 cells from 4 utricles), AAV-Stk32a [(K52R)]-P2A-EGFP at P0 (N=108 cells from 4 utricles). (**E–G**) Hair cells transduced with virus expressing STK32A::EGFP (**G,G′**) fusion show EGFP immunofluorescence throughout the stereociliary bundle (arrow) that overlaps with phalloidin (**H,H′**) and at the apical surface (arrowhead). Hair cells transduced with STK32A [Δ2G]::EGFP lacking N-myristoylation does not colocalize with phalloidin in the apical compartment or bundle (**G′,H′,I′,J′**) Cellular profiles extracted from image stacks corresponding to G,H,I&J. Scale bars: A-C (10 μm), E (5 μm).

The online version of this article includes the following figure supplement(s) for figure 3:

**Figure supplement 1.** STK32A::EGFP (C-terminal fusion) is functionally active.

with kinocilium that were located on one side of the apical cell surface adjacent to the tallest row of stereocilia (*Figure 5A–B*).

This *Stk32a* [-/-] phenotype resembles that of PCP mutants in which hair cells are also misoriented relative to their neighbors (*Stoller et al., 2018*; *Yin et al., 2012*). However, immunofluorescent labeling of core PCP proteins suggests that PCP signaling remains intact in *Stk32a* [-/-] sensory epithelia. For example PK2, which is asymmetrically distributed along one side of the hair cell (*Deans et al., 2007*), remains normally distributed in *Stk32a* [-/-] hair cells (*Figure 5C–D*). As a result, in some *Stk32a* [-/-] hair cells, PK2 can be found on the side of the bundle rather than adjacent to or opposite of the kinocilium as seen in controls. In addition, the polarized distribution of CELSR1 at junctions between supporting cells is not altered in *Stk32a* [-/-] (*Figure 5E, F*). *Emx2* expression in the lateral extrastriolar region of the utricle and adjacent transitional epithelia was also not impacted by *Stk32a* gene deletion (*Figure 5G and H*) consistent with this transcription factor acting upstream of *Stk32a* expression. Together these observations demonstrate an STK32A function in the medial utricle that coordinates intrinsic polarity of the hair cells with the PCP axis, and that is inhibited in the lateral utricle by EMX2.

## STK32A regulates GPR156 distribution at apical cell boundaries

In the lateral region of the utricle and inner region of the saccule, EMX2 acts upstream of the G-protein-coupled receptor GPR156 to reorient stereociliary bundles and thereby position the LPR (*Kindt et al., 2021*). Remarkably, GPR156 function is regulated post-translationally because it is expressed by all vestibular hair cells, but can only be detected at the apical hair cell surface in hair cells that also express *Emx2*. The mechanisms regulating GPR156 distribution at the cell surface are not known, although it seems likely that factors transcriptionally regulated by EMX2 contribute to this process.

The potential for STK32A to regulate GPR156 distribution at the apical surface of hair cells was tested by visualizing GPR156 protein in the *Stk32a* [-/-] maculae and semi-circular canal cristae (*Figure 6*). For these experiments, Pericentrin immunolabeling was used to mark the basal body and determine hair cell orientation. In controls, GPR156 was polarized apically in hair cells of the lateral utricle and inner saccule (*Emx2*-positive domains), but not in the rest of the organs (*Emx2*-negative domains), as reported previously (*Figure 6A and C*). In *Stk32a* [-/-], however, GPR156 was also detected apically in the *Emx2*-negative domains of the utricle (*Figure 6B*) and saccule (*Figure 6D*). Similarly, GPR156 was detected at the cell boundary in *Stk32a* [-/-] cristae, but not in control cristae as expected (*Figure 6E and F* and *Kindt et al., 2021*). Despite ectopic localization at apical cell boundaries in *Emx2*-negative hair cells, GPR156 was not tightly polarized as seen in control utricular cells on the lateral side of the LPR, and was either symmetrical and surrounded individual cells, or was polarized but imprecisely aligned with the PCP axis (*Figure 6B″, D″ and F″*).

To directly test whether STK32A negatively regulates GPR156 delivery to or retention at the cell surface, *Stk32a-P2A-EGFP* plasmids were electroporated into auditory hair cells at E13.5 and the cochlea cultured for an additional week (*Figure 7*). Auditory hair cells were selected for experimentation because plasmid electroporation is more efficient and GPR156 enrichment is easier to detect than for vestibular hair cells (*Kindt et al., 2021*). *Stk32a* is also not normally expressed in the auditory

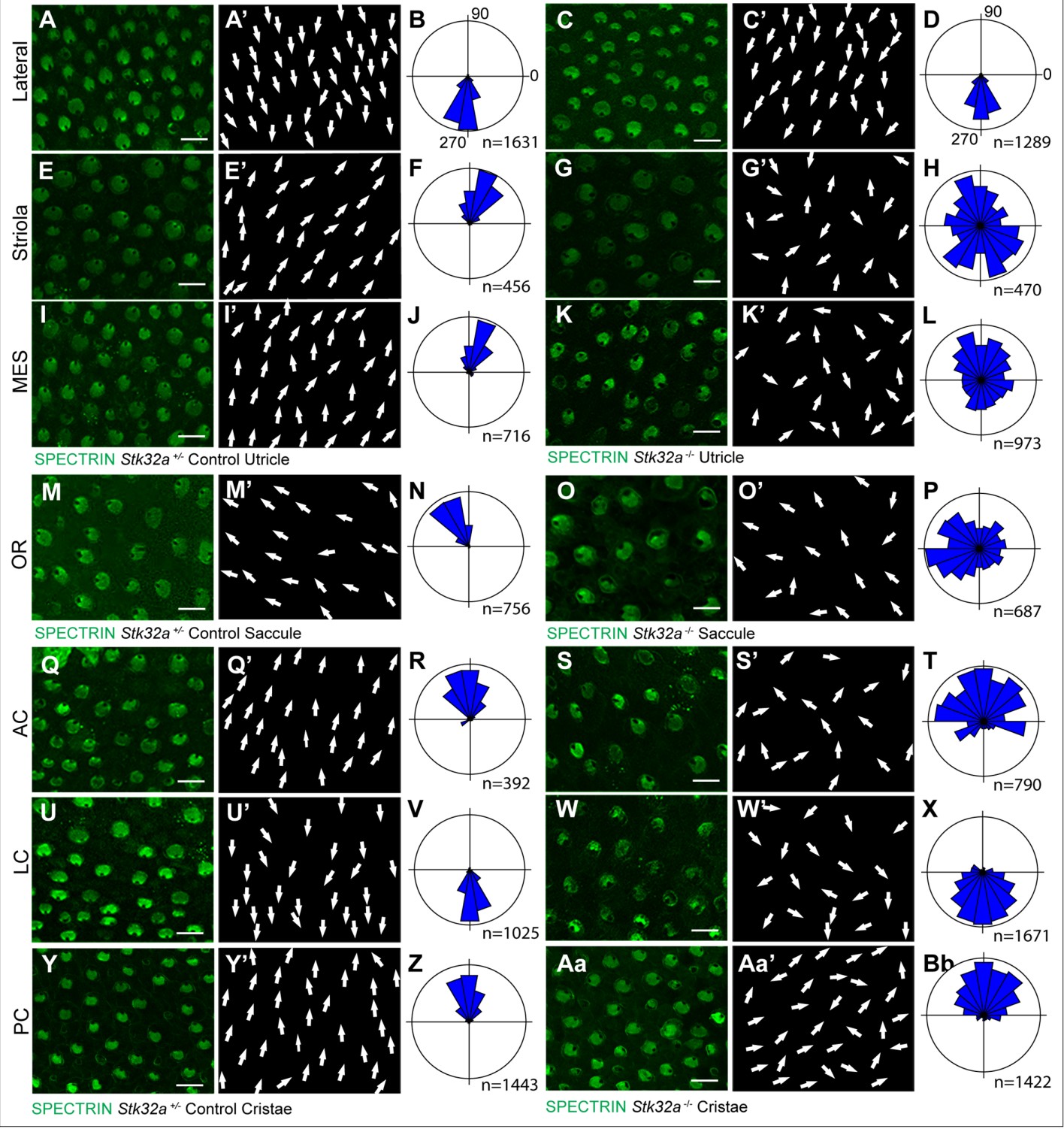

**Figure 4.** STK32A coordinates stereociliary bundle orientation with the PCP axis. Stereociliary bundle orientation in inner ear sensory epithelia from P5 heterozygous control and *Stk32a* ⁻/⁻ mice determined by labeling the cuticular plate with antibodies against βII-SPECTRIN. (A,A',C,C') Hair cells in the lateral region of the utricle retained normal bundle orientations in both genotypes. (E,E',G,G') Hair cells in the *Stk32a* ⁻/⁻ striolar region appear randomly oriented relative to controls. (I,I',K,K') Hair cells in the *Stk32a* ⁻/⁻ medial extrastriolar region (MES) appear randomly oriented relative to controls. (M,M',O,O') Hair cells in the outer region of the *Stk32a* ⁻/⁻ saccule appear disorganized relative to controls. (Q,Q',U,U',Y,Y') Hair cells in cristae of controls are tightly aligned with the associated semi-circular canal while (S,S',W,W',Aa,Aa') *Stk32a* ⁻/⁻ cristae contain disorganized hair cell bundles. Cumulative graphs of individual bundle orientations for each sensory epithelia pooled from multiple littermate controls (B,F,J,N,R,V,Z) or *Stk32a* ⁻/⁻ (D,H,L,P,T,X,Bb)

*Figure 4 continued on next page*

*Figure 4 continued*

(N=3–4 sensory organs for each). Summary of analysis field locations is illustrated in *Figure 4—figure supplement 1*. (MES) medial extrastriolar, (OR) outer region, (AC) anterior cristae, (LC) lateral cristae, (PC) posterior cristae. Scale bars: 10 µm.

The online version of this article includes the following source data and figure supplement(s) for figure 4:

**Figure supplement 1.** *Stk32a* gene targeting strategy and validation.

**Figure supplement 1—source data 1.** DNA electrophoresis of PCR products.

**Figure supplement 2.** *Emx2* and *Stk32a* expression domains and planar polarity analysis fields.

**Figure supplement 3.** *Stk32a*⁺ᐟ⁻ mice do not display planar polarity phenotypes.

**Figure supplement 4.** Auditory hair cells are not impacted by *Stk32a* gene deletion.

epithelium. Individual hair cells ectopically expressing *Stk32a* had decreased levels of GPR156 or displayed no GPR156 protein at their apical cell surface (*Figure 7A, A', B and B'*). This reduction was quantified by calculating a ratio of GPR156 enrichment at the medial cell boundary of electroporated versus non-electroporated neighbors for *Egfp* control and *Stk32a* constructs (*Figure 7E*). In addition to changes in GPR156, a subset of *Stk32a-P2A-EGFP* expressing hair cells (8 of 50) also had an inverted intrinsic planar polarity that was revealed by changes in the distribution of aPKC (*Figure 7C and C'*). In contrast, auditory hair cells electroporated with constructs expressing the predicted kinase-dead *Stk32a* $^{K52R}$ mutant continued to display GPR156 protein at their apical cell boundaries (*Figure 7D and D'*) suggesting that STK32A kinase activity is required for the negative regulation of GPR156 distribution. Notably, the asymmetric distribution of GPR156 was also impacted by the *Stk32a* $^{K52R}$ mutation because GPR156 could be detected at both the medial and lateral cell boundaries (*Figure 7D and F*). While the molecular basis of this change is not known, it is possible that physical interactions between STK32A and GPR156 underlie the redistribution of GPR156 following the overexpression of STK32A $^{K52R}$ throughout the hair cell. Altogether, these gene deletion and electroporation-mediated overexpression assays demonstrate that STK32A is required for the post-translational regulation of GPR156 function through a phosphorylation-dependent event that regulates the presence of this receptor at the apical hair cell surface.

## Discussion

### Summary and model

We have identified the serine-threonine kinase STK32A as a planar polarity factor that functions opposite of EMX2 and provide genetic evidence that STK32A is necessary and sufficient to determine the orientation of mouse vestibular hair cells. *Stk32a* transcription is repressed in the lateral utricle and inner region of the saccule and in the absence of *Emx2*, *Stk32a* expression is expanded to include all hair cells in those regions. This expansion likely explains how in the *Emx2* mutant, lateral hair cells in the utricle remain uniformly oriented along the PCP axis albeit with bundle orientations characteristic of medial hair cells (*Jiang et al., 2017*). We further demonstrate that in the absence of *Stk32a*, medial hair cells are randomly oriented and no longer aligned with the underlying PCP axis. Finally, we demonstrate that STK32A contributes to LPR formation by regulating the sub-cellular distribution of GPR156 so that it is only present at the apical boundaries of hair cells where EMX2 is expressed and *Stk32a* transcription is repressed. Based upon these observations we propose the following model for how STK32A functions together with EMX2 and GPR156 to pattern hair cells about the LPR (*Figure 8*). In this model, GPR156 functions at the apical cell boundaries to align stereociliary bundles with the underlying PCP axis so that the basal body is positioned adjacent to PK2 in EMX2-positive regions of the utricle and saccule (*Jiang et al., 2017*; *Kindt et al., 2021*). STK32A serves a similar function in EMX2-negative regions by positioning the basal body opposite of PK2 though the targets of STK32A kinase activity at this step remain unknown. The LPR is formed along the boundary of *Emx2* expression and is maintained by hair cells through two steps of negative regulation. In the first, STK32A inhibits GPR156 delivery or retention at the apical boundary of cells located on the medial, EMX2-negative side of the LPR. In the second, EMX2 represses *Stk32a* transcription in those hair cells located on the EMX2-positive side of the LPR, allowing GPR156 to accumulate at the cell surface where it can promote Emx2-dependent bundle reversal. An essential tenet of this model is that intercellular PCP signaling functions solely to align neighboring cells along a common polarity axis and is not impacted

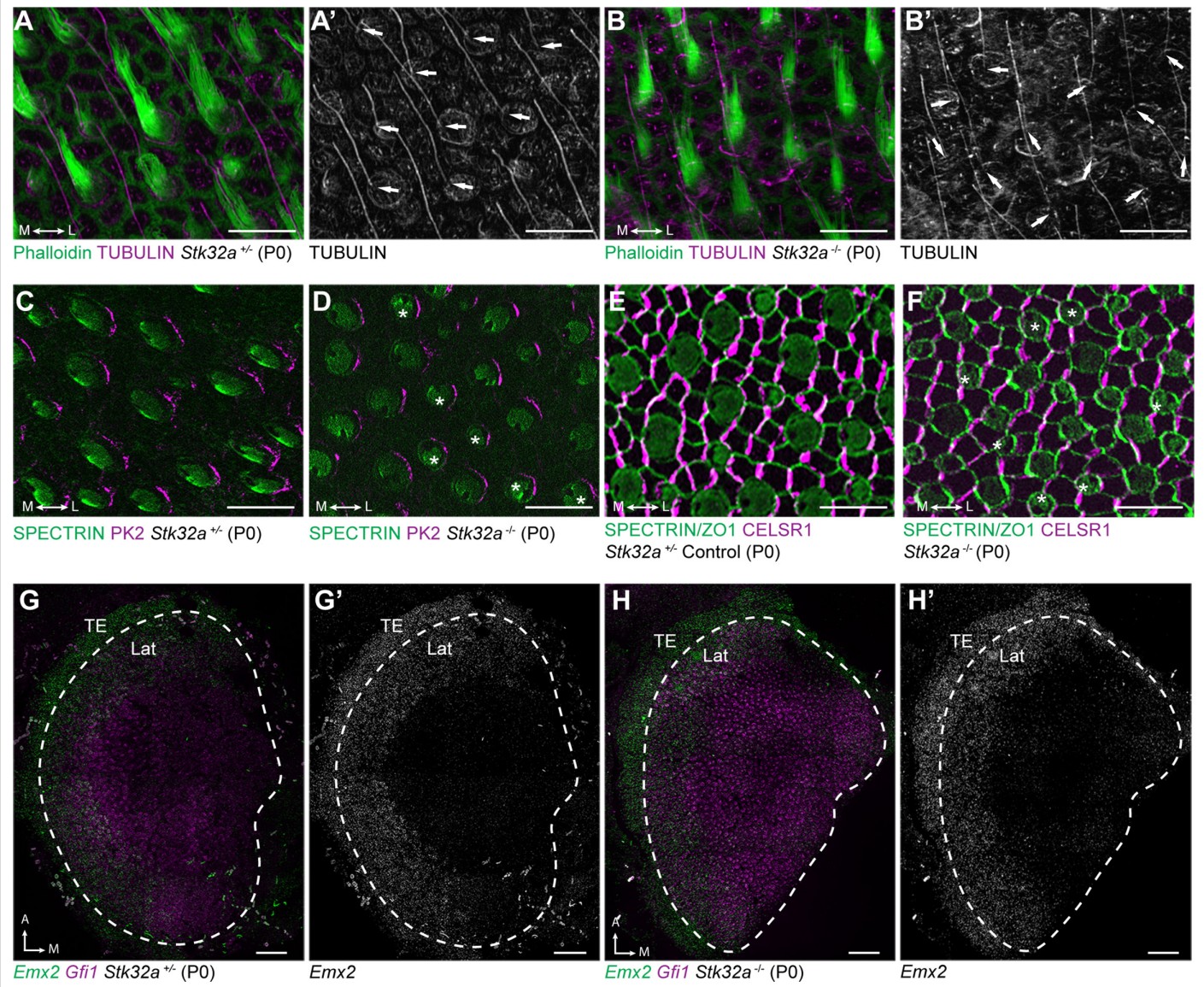

**Figure 5.** Stereociliary bundle polarization, PCP and LPR patterning remains intact in *Stk32a* mutants. (**A**) P0 heterozygous control and (**B**) *Stk32a* [−/−] stereociliary bundles immunolabeled with acetylated-tubulin antibodies to mark the kinocilium and phalloidin to label the stereocilia. *Stk32a* [−/−] hair cells from intact polarized bundles despite being misoriented. Arrows indicated bundle orientation. (**C,D**) PK2 is asymmetrically distributed along one side of vestibular hair cells from the striolar region of control (**C**) and *Stk32a* [−/−] (**D**) utricles. Asterisks mark examples of misoriented cells where the fonticulus is not correctly positioned opposite of PK2 in the *Stk32a* [−/−]. (**E,F**) CELSR1 immunolabeling at junctions between supporting cells reveals the orientation of the PCP polarity axis in heterozygous control (**E**) and *Stk32a* [−/−] (**F**) utricles. Combined labeling of βII-SPECTRIN and ZO-1 reveals hair cell orientation and junctions between supporting cells. Asterisks mark examples of misoriented hair cells in *Stk32a* [−/−]. (**G,G'**) fISH for *Emx2* and *Gfi1* demonstrating *Emx2* expression in the lateral utricle and adjacent transitional epithelia (TE) of littermate controls. Dashed lines demarcate the boundaries of *Gfi1*-expressing hair cells. (**H,H'**) In *Stk32a* [−/−] utricles, the *Emx2* expression domain is not changed relative to *Gfi1*. Scale bars:A-F (10 µm); G,H (50 µm).

by *Emx2*, *Gpr156* or *Stk32a* mutations as has been shown previously (*Jiang et al., 2017*; *Kindt et al., 2021* and *Figure 5*). In this model, the final position of the basal body and kinocilium relative to the core PCP proteins are determined by STK32A or GPR156.

Yet to be determined is whether EMX2 acts directly as a transcriptional repressor that binds the *Stk32a* promoter or indirectly through intermediary factor(s) to regulate *Stk32a* expression. Prior transcriptional profiling has shown that EMX2 contributes to both transcriptional activation and repression in the developing telencephalon prior to cortical arealization (*Hamasaki et al., 2004*; *Li et al., 2006*), but these studies also did not distinguish between direct or indirect mechanisms. DNA binding assays

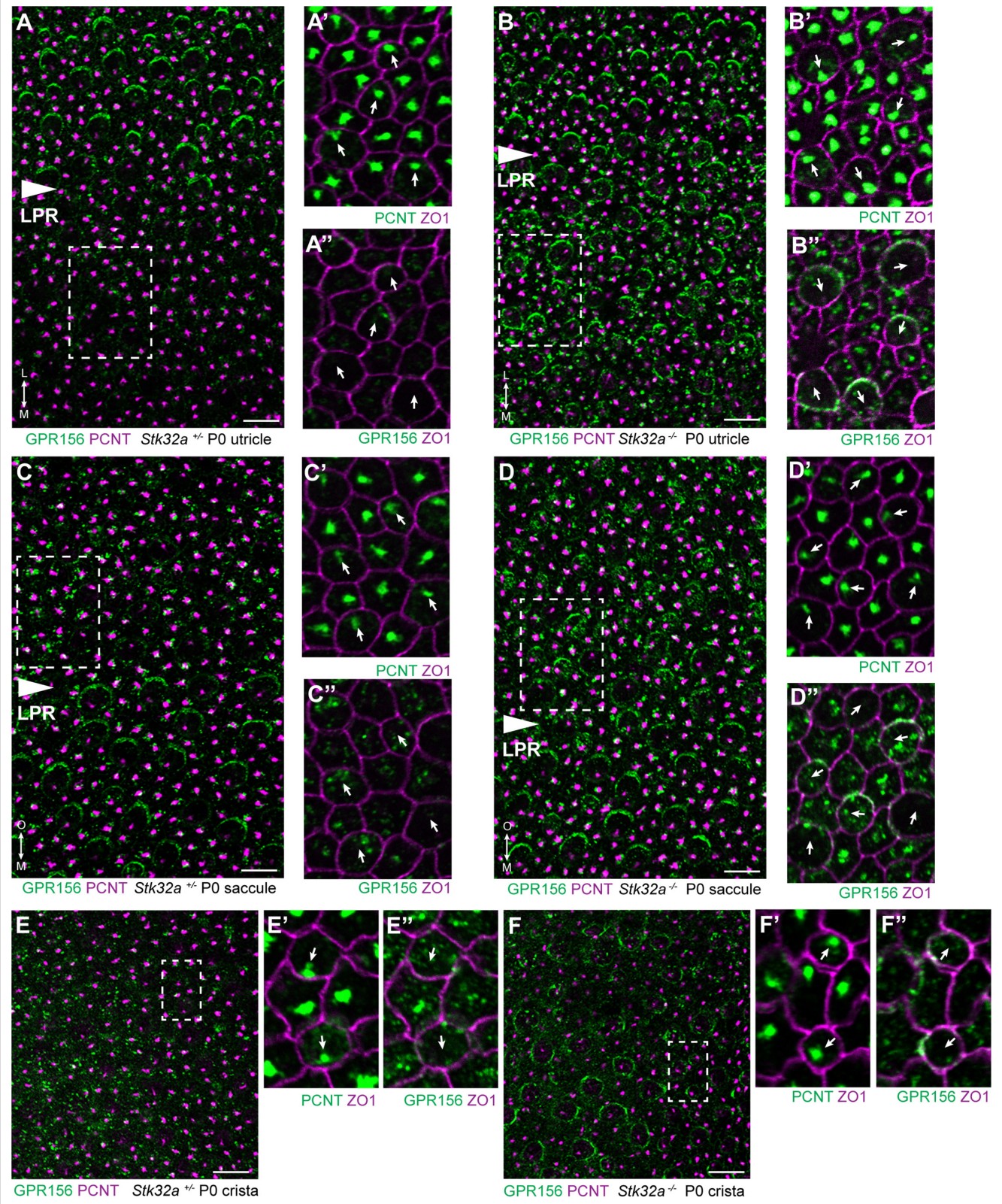

**Figure 6.** STK32A regulates GPR156 distribution at the apical cell surface. (**A**) GPR156 is asymmetrically distributed at apical cell boundaries of hair cells in the lateral region of the utricle along one side of the LPR. Pericentrin (PCNT) shows the position of the basal body beneath cilia in hair cells and supporting cells. (**A'**) Higher magnification image of the boxed region from 'A' taken from the medial side of the LPR. Pericentrin and ZO-1 labeling reveals the position of the basal body and hence orientation of the stereociliary bundle (arrows). (**A''**) The distribution of GPR156 relative to ZO1 in the

*Figure 6 continued on next page*

*Figure 6 continued*

boxed region from 'A'. (**B-B"**) In *Stk32a* [-/-] utricles, GPR156 is redistributed to the apical boundaries of hair cells on the medial side of the LPR, although the uniform asymmetric distribution is lost. (**C-C"**) GPR156 is asymmetrically distributed at apical cell boundaries of hair cells in the inner region of the saccule along one side of the LPR. (**D-D"**) GPR156 is similarly redistributed and can be detected at the apical boundaries of hair cells on either side of the LPR in the *Stk32a* [-/-] saccule. For these experiments the position of the LPR was determined based upon the organization of stereociliary bundles which are aligned in the Emx2-positive region and misoriented in the Emx2-negative region in *Stk32a* [-/-]. (**E-E"**) In littermate control cristae GPR156 signal is visible but not at apical cell boundaries. (**F-F"**) In *Stk32a* [-/-] cristae, GPR156 is redistributed to the apical cell boundaries, although similar to the utricle and saccule GPR156 distribution is not uniformely planar polarized. Scale bars: 10 µm.

have shown that EMX2 is capable of occupying POU sites in the *Sox2* enhancer and repressing *Sox2* expression in a dose-dependent manner (*Mariani et al., 2012*). The positive and negative regulation of gene expression may also be determined by cofactors since EMX2 represses *Gsx2* gene expression during earlier stages of telecephalic development but only in the presence of DMRT3 and DMRT5 (*Desmaris et al., 2018*). Overall the precedent set by these prior studies of telencephalic and cortical development is consistent with our interpretation and that EMX2 contributes to transcriptional repression in the vestibular sensory organs. Moreover, they highlight the diversity of mechanisms that might mediate this effect including the potential for additional unidentified intermediaries or co-factors specific to vestibular hair cells.

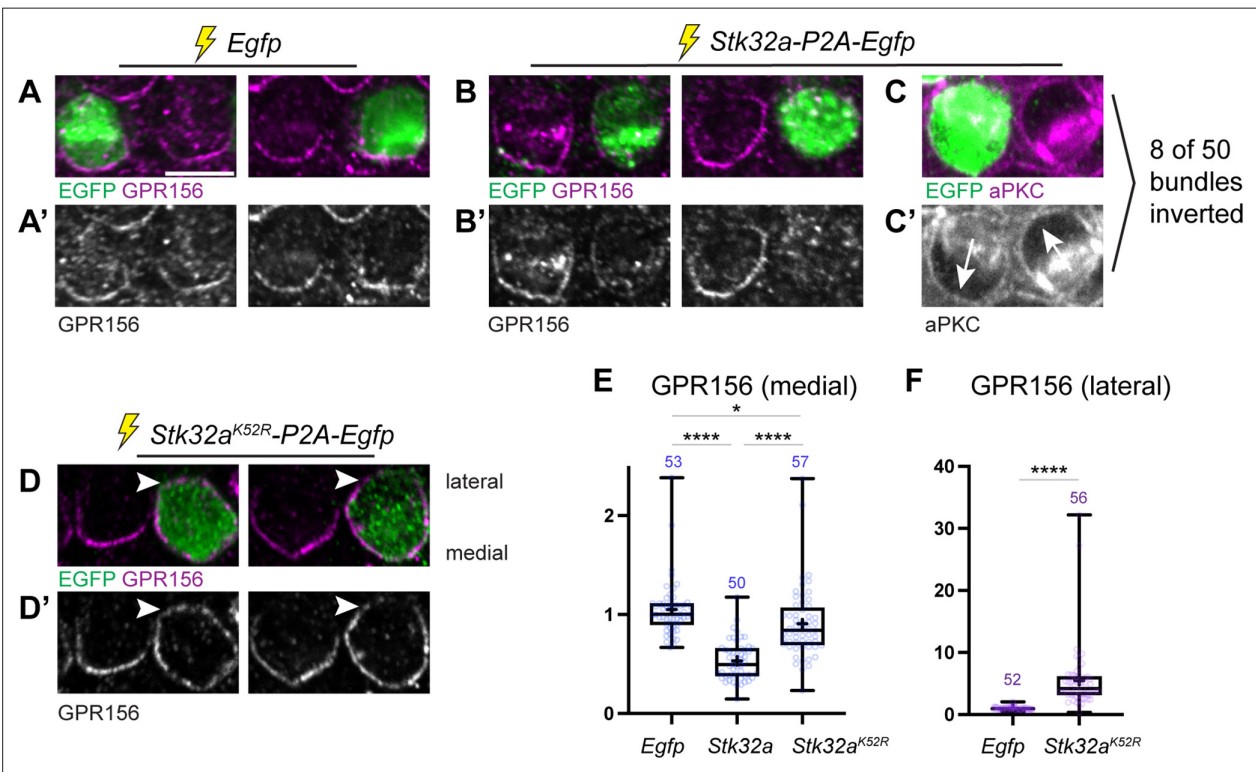

**Figure 7.** Ectopic expression of STK32A in auditory hair cells disrupts GPR156 localization. (**A,A'**) Cochlea electroporated at E13.5 and grown in culture develop auditory hair cells with distinct crescents of GPR156 enriched at their medial cell boundaries. Two examples of an EGFP-positive electroporated cell with a non-electroporated neighbor are provided. (**B,B'**) When electroporated with *Stk32a-P2A-EGFP*, hair cells expressing the EGFP reporter have a stark reduction in GRP156 at their apical cell boundaries. (**C,C'**) A minority of cells expressing STK32A (8/50, 16%) had reversed planar polarity as revealed by the distribution of the intrinsic polarity marker aPKC. Arrows indicated bundle orientations. (**D,D'**) GPR156 distribution at the medial cell boundary of hair cells expressing the predicted kinase-dead STK32A [K52R] mutant was only mildly impacted while GPR156 distribution at the lateral boundary of these cells increased (arrowheads). (**E**) The intensity of GPR156 immunolabel fluorescence at medial cell boundaries in electroporated relative to neighboring cells is significantly reduced in cells that ectopically express STK32A and only modestly impacted in cells expressing STK32A [K52R]. Pairwise comparisons of relative fluorescence intensity were evaluated by one-way Anova (**** p<0.0001, * p=0.0127). (**F**) The intensity of GPR156 immunolabel fluorescence at lateral cell boundaries in electroporated relative to neighboring cells is significantly increased in cells that ectopically express STK32A [K52R]. Pairwise comparisons of relative fluorescence intensity were evaluated using the Mann-Whitney test (**** p<0.0001). Gpr156 data is graphed as 25-75% boxplots where exterior lines show minimum and maximum values, the middle line the median and + the mean. Scale bars: 5 µm.

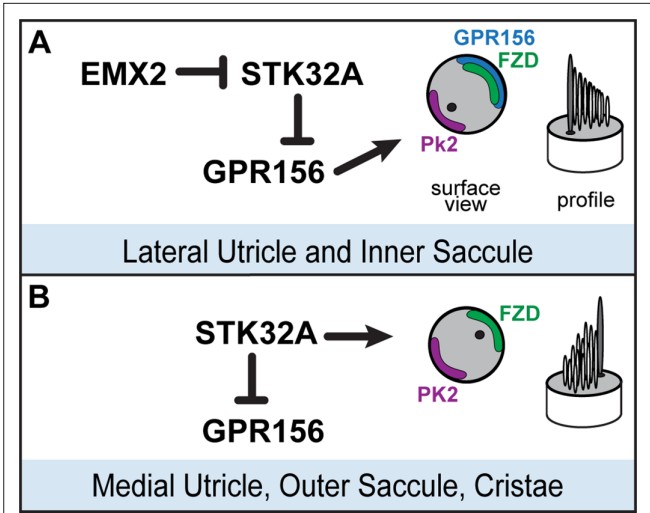

**Figure 8.** Model for patterning hair cells about the LPR. (**A**) In the lateral region of the utricle and inner region of the saccule, GPR156 functions to orient the stereociliary bundle relative to the PCP axis so that the basal body and kinocilium are positioned adjacent to PK2. Repression of STK32A transcription by EMX2 enables GPR156 function and establishes the position of the LPR. (**B**) In the medial utricle, outer region of the saccule and the semi-circular canal cristae, STK32A orients the stereociliary bundle so that the basal body and kinocilium are positioned adjacent to FZD. Although the direct substrates of STK32A in these regions are not known, STK32A-dependent regulation of GPR156 at apical cell boundaries prevents orientation reversal and reinforces the position of the LPR.

## Single or dual functions for STK32A?

Based upon our analysis of *Stk32a* knockouts one possibility could be that that STK32A has two functions during vestibular hair cell development. The appearance of randomly oriented bundles in the medial region of the mutant utricle and outer region of the saccule suggests that one function is to align bundle orientation with the PCP axis. Thus, in *Stk32a* $^{-/-}$ hair cells the basal body either moves incorrectly relative to the PCP axis or is not maintained in a fixed position following an initially correct movement. A second function is revealed by the ectopic overexpression of STK32A in cochlear hair cells where the apical distribution of GPR156 is similar to EMX2-positive vestibular hair cells. Although not normally expressed in these cells, the addition of STK32A dramatically reduces that amount of GPR156 detected at apical cell boundaries. Together with the aberrant appearance of GPR156 at the surface of EMX2-negative vestibular hair cells in *Stk32a* $^{-/-}$ mice, these data demonstrate a second STK32A function in the post-translational regulation of GPR156. STK32A could do this either by regulating a key step of GPR156 trafficking and delivery to the cell surface, or by activating the internalization of GPR156 from the cell surface after delivery has occurred. It should be noted however, that the distribution the GPR156 in *Stk32a* $^{-/-}$ hair cells on the EMX2-negative side of the LPR does not have the same uniform planar polarized distribution that characterizes GPR156 distribution in hair cells that have EMX2. This suggests the contributions of additional, yet to be identified, factors that also regulate GPR156 localization. The disorganized distribution of GPR156 also raises an alternative possibility that STK32A only has one function during hair cell development which is to negatively regulate GPR156. In this scenario, the misoriented stereociliary bundles seen in *Stk32a* $^{-/-}$ would not be due to an STK32A function in regulating the position of the kinocilium. Instead the disorganization could be an artifact of GPR156 which does not normally function in these cells, is lacking a planar polarized organization, and could therefore disrupt the proper positioning of the kinocilium. This simplified role of STK32A would not disrupt the general tenets of the model we propose for EMX2, STK32A and GPR156 function (*Figure 8*), and in this scenario we would propose that the core PCP proteins determine the position of the kinocilium in hair cells not expressing EMX2.

## STK32A localization and substrates

A limitation to the current study is that we lack a suitable antibody for evaluating the subcellular distribution of STK32A which limits the detail of the mechanistic models that can be proposed.

Predictions based on sequence analysis suggest that SKT32A is membrane-associated due to the potential for myristoylation of an N-terminal glycine (*UniProt Consortium, 2021*; *UniProt C, 2023*). Our observations that EGFP-tagged STK32A is present in stereocilia and at the apical cell surface and that this localization is lost following deletion of the N-terminal glycine, is consistent with myristoylation-dependent targeting to this location. This apical localization of STK32A would be most consistent with an STK32A function to promote the internalization of GPR156 from the apical cell boundary rather than trafficking and delivery to that location. Alternatively, STK32A has been predicted to localize to cilia (*Atlas THP, 2022*; *Thul et al., 2017*) and therefore may be aptly suited to position the kinocilium relative to the PCP axis. Consistent with this alternative, a large-scale immuno-precipitation and mass spectrometry analysis of kinase binding partners found that STK32A associated with components of the chaperonin-containing T-complex (TRiC) (*Buljan et al., 2020*). This complex may play a role in formation of the BBSome, a complex associated with cilio-genesis (*Seo et al., 2010*) that has also been associated with ciliopathies and planar cell polarity (*Ross et al., 2005*). A better understanding of STK32A function will occur following the identification of the phosphorylation substrates and potential binding partners for this kinase. A striking candidate is GPR156 since we show that STK32A can negatively regulate its localization to the cell surface. Despite this we have been unable to demonstrate physical interactions or phosphorylation events for these two proteins following overexpression in heterologous cell lines, raising the possibility that GPR156 regulation is indirect and requires additional, intermediary proteins also found in hair cells.

## STK32A as a dark kinase

The human genome encodes over 500 kinases yet despite their identification and classification, the substrates and signaling pathways for approximately 160 of these kinases remain poorly understood (*Berginski et al., 2021*). These remaining enzymes have been dubbed Dark Kinases and include the small protein family of STK32A, STK32B, and STK32C. To the best of our knowledge, no prior studies have associated the STK32/YANK kinase family with hair cells, planar polarity or inner ear development. Based on sequence homology and crystal structure the STK32/YANK family is a sub-group of the ACG kinases which include the better understood PKA, PKC and PKG proteins (*Sorrell et al., 2020*). Although prior knowledge of STK32A is limited, it has been linked to stomach adenocarcinoma (STAD) where its overexpression is associated with lower overall survival (*Southekal et al., 2021*), and somatic mutations in *Stk32a* have also been identified in neuroendocrine carcinoma of the lung and metastatic melanoma (*Greenman et al., 2007*). Our analyses reveal that this kinase is a regulator of cellular polarity that aligns individual cells with the planar polarity axis found in many tissues and epithelia. This is an interesting observation because disrupted planar cell polarity signaling is also associated with cancer progression and metastasis (*VanderVorst et al., 2018*; *Wang, 2009*). An exciting prospect is that pharmacologic manipulation of STK32 family kinases function or their substrates could be developed as a treatment to slow progression. Towards this end, small molecules that bind to STK32A have been identified which may be useful for further research (*Sorrell et al., 2020*).

## Emx2 and hair cell patterning about the LPR

The results presented here remain consistent with EMX2 acting as a master regulator or switch controlling stereociliary bundle orientation and thereby determining the position of the LPR. However, our RNAseq analyses demonstrate that EMX2 does not establish the LPR through transcriptional activation but rather via the repression of genes such as *Stk32a*. Consistent with this, the genes that we identified as being positively regulated by EMX2 were predominantly expressed in the transitional epithelia rather than vestibular hair cells. Based on these findings we propose that EMX2 functions more broadly to define the regional identity of the lateral utricle, inner region of the saccule and neighboring non-sensory epithelia. As such EMX2 would sit atop a gene regulatory network that regulates multiple differentiation steps with hair cell planar polarity and patterning about the LPR just one of these events. We provide evidence that a key component enacting this EMX2 planar polarity effect in hair cells is the dark kinase STK32A.

## Materials and methods

### Mouse strains and husbandry

*Emx2* $^{Cre/wt}$ mice (**Kimura et al., 2005**) were provided by S.Aizawa (Riken Institute), maintained by backcross with B6129SF1/J females (Jax strain#101043) and *Emx2* mutants generated by intercrossing *Emx2* $^{Cre/wt}$ male and female mice. Tissue from *Gfi1*-Cre, *Rosa26* $^{LSL-Emx2/wt}$ transgenic mice and *Rosa26* $^{LSL-Emx2/wt}$ controls were provided by Doris Wu (NIH/NIDCD). The *Stk32a* KO mouse line was generated by targeting the second exon of the *Stk32a* gene, which contains the translational start, using CRISPR technology. Two gRNAs (5'-UAGAAAUGACCAUGGUGCAU-3' and 5'-UACA GUGUAUCUAAGGAUAG-3') that are complementary to intronic DNA sequences located on either side of *Stk32a* exon 2 were designed by the University of Utah Mutation Generation and Detection Core. These were combined with recombinant Cas9 protein for pronuclear injections into B6CBAF1/J mouse zygotes (Jax strain#100011) that were subsequently implanted into pseudo-pregnant females by University of Utah Transgenic and Gene Targeting Core. One male founder with an appropriate deletion generated by NHEJ and identified by PCR amplification followed by Sanger sequencing was selected for line expansion through backcross with B6129SF1/J females. PCR primers used for mutant genotyping are: *Stk32a* primer A (5'-AGTTTTCTTGATACCTGGGGATGT-'3') and *Stk32a* primer B (5'-GGATGGCAGTGATTATGAAAGTTG-3'). The *Stk32a* $^{-/-}$ phenotype was initially evaluated by intercrossing N2 heterozygous mice and the mutant phenotype was consistently observed when N3 and later generations were experimentally intercrossed. Experimental crosses consisted of *Stk32a* $^{-/-}$ males crossed with *Stk32a* $^{+/-}$ females, and heterozygous littermates generated from these crosses were used as controls. CD1 mice were purchased from Charles River and intercrossed for utricle explant experiments. Mice were genotyped by PCR using allele specific primers. All animal work was reviewed for compliance and approved by the Animal Care and Use Committee of The University of Utah (IACUC protocol #00001498) and the Animal Care and Use Committees of The Jackson Laboratory (Animal Use Summary AUS no. 14012).

### Antibodies and wholemount immunolabeling

Inner ear tissues were fixed using 4% paraformaldehyde in 67 mM Sorensons' phosphate buffer (pH 7.4) for 2 hr on ice, or using 10% trichloroacetic acid (TCA) for 10 min on ice depending upon primary antibody selections (see below). For wholemount immunofluorescent labeling, utricles and saccules were micro-dissected and detergent permeabilized using a blocking solution (5% normal donkey serum, 1% bovine serum albumin (BSA), PBS) supplemented with 0.5% Triton-X100. Primary antibodies and Phalloidin Alexa Fluor 488 (Invitrogen A12379) were diluted in blocking solution supplemented with 0.1% Tween-20 and incubated overnight at 4 °C. Tissue was washed thoroughly using PBS with 0.05% Tween-20 and incubated with species-specific Alexa-conjugated or Cy3 conjugated secondary antibodies (Jackson ImmunoResearch, 705-165-147, 705-605-147, 706-165-148, 711-165-152, 711-545-152, 711-605-152, 712-605-150), before mounting for fluorescence microscopy using Prolong Gold (ThermoFisher Scientific, P36930). Fluorescent images were acquired by structured illumination microscopy using the Zeiss Axio Imager M.2 with ApoTome.2 attachment and Axiocam 506 mono camera. Images in *Figures 6 and 7* (Gpr156 experiments) were acquired using a Zeiss LSM800 confocal microscope and a 63 x NA1.4 objective. Images were processed using Zeiss Zen software and figures were prepared using with Adobe Illustrator.

Antibodies and fixatives used in this study are: Mouse anti-Acetylated Tubulin (1:250, PFA, Sigma-Aldrich T7451, RRID:AB_609894), Mouse anti-aPKC (1:100, TCA, Santa Cruz Biotechnology SC-17781), Mouse anti-βII Spectrin (1:1000, PFA or TCA, BD Biosciences 612562, RRID:AB_399853), Guinea Pig anti-CELSR1 (1:500, TCA, F.Tissir (Louvain)), Rabbit anti-GFP (1:1000, PFA, Life Technologies A11122, RRID:AB_221569), Goat anti-GPR156 (1:100, TCA, Santa Cruz Biotechnology SC-102572, RRID:AB_10839485), Rabbit anti-MST4/STK26 (1:500, PFA, Abcam Ab52491, RRID:AB_881249), Goat anti-Oncomodulin/OCN (1:250, PFA, Santa Cruz Biotechnology SC-7466, RRID:AB_2267583), Rabbit anti-pericentrin/PCNT (1:400, TCA, Biolegend PRB-432C, RRID:AB_291635), Rabbit anti-PK2 (1:200, PFA [*Deans et al., 2007*]), Rat anti-ZO1 (1:1000, TCA, DSHB R26.4C, RRID:AB_2205518).

### RNA sequencing

Whole inner ear capsules were dissected from E18.5 mouse pups, generated by *Emx2* $^{Cre/wt}$ intercross, under nuclease free conditions. The bony labyrinth was lightly perforated and ears were stored in

RNAlater (ThermoFisher AM7020) at 4 C for 1 week to allow for subsequent collections and genotyping. *Emx2* *Cre/Cre* and wild type samples were dissected under nuclease free conditions to isolated the utricle from surrounding mesenchyme and adjacent cristae. Four to 6 utricles were combined per genotype to constitute one biological replicate, and 4 replicates were collected for each. Tissue was transferred to 350 µl of RNeasy Lysis buffer (Qiagen 79216) Buffer RLT and lysed using the Qiagen Tissue Lyser LT at 20 Hz with one stainless steel bead for 4 min. Following centrifugation, supernatant was transferred to a clean tube and RNA isolated using the RNeasy Micro Kit (Qiagen 74004). DNA and RNA concentrations were determined using Qubit broad range DNA and RNA assay kits (Invitrogen Q32853 Q10211) and RNA quality evaluated using the Agilent 2200 TapeStation which returned RNA integrity numbers ranging from 8.3 to 9.1. Libraries were generated at the Huntsman Cancer Institute Genomic Core facility using TruSeq Stranded Total RNA Library Prep with Ribo-Zero Gold kits (Illumina 20020598) and sequenced using the Illumina NovaSeq 6000 for 25 M 2x50 bp reads per biological replicate. Significant unmapped reads were found in one WT sample which was excluded from subsequent analysis after principal component analysis revealed it to be a significant outlier. Remaining samples were compared using the Bioconductor DESeq package as well as by post hoc-analysis using Swish. Benjamini-Hirshberg false discover rate was used to normalize data across all transcripts and produce adjusted p-values for significantly downregulated or upregulated genes based on log2-fold changes compared to WT baseline transcript levels.

## Fluorescent In Situ Hybridization (fISH) by Hairpin Chain Reaction (HCR v3.0)

For wholemount fISH inner ears were fixed overnight at 4 °C using 4% paraformaldehyde prepared in phosphate buffered saline (PBS, pH 7.4) and stored as needed at –20 °C in 100% MeOH. Prior to hybridization tissue was dissected to expose sensory epithelia and rehydrated through a decreasing MeOH gradient (75%, 50%, 25%) ending in PBST (PBS with 0.1% Tween-20). fISH was completed using Hairpin Chain Reaction protocols (*Choi et al., 2018*). In brief, samples were digested in 30 µg/ml proteinase K for 20 min at room temperature followed by post-fixation in 4% PFA/PBS for 20 min. After rinsing with PBST, samples were equilibrated in hybridization buffer (30% formamide, 5 X sodium chloride sodium citrate (SSC), 0.9 mM citric acid (pH 6.0), 50 µg/mL heparin, 1 X Denhardt's solution, 10% low MW dextran sulfate) at 37 °C for 30 min, then incubated in hybridization buffer containing 16 nmol mRNA-specific probes overnight at 37 °C. Excess probe were removed using 4 exchanges of probe wash buffer (30% formamide, 5XSSC, 9 mM citric acid (pH 6.0), 0.1% Tween 20, 50 µg/mL heparin) followed by2 exchanges of 5XSSCT (0.1% Tween in 5XSSC), then fluorescent signal amplification was performed by incubating samples with hairpin mixtures in amplification buffer (5XSSC, 0.1% Tween, 10% low MW dextran sulfate) overnight at room temperature. Hairpins were removed by extensively washing with 5XSSCT, and samples were transferred to glass slides and coverslipped using ProLong Gold Antifade mounting media (Invitrogen). Fluorescent images were captured by structured illumination microscopy using a Zeiss Axio Imager M.2 with ApoTome.2 attachments and an Axiocam 506 monochrome camera. HCR probes used in this study were: Caln1-B1 (Molecular Instruments PRF994), Emx2-B5 (Molecular Instruments PRA862), Gfi1-B2 (Molecular Instruments PRG075), Slc26a4-B1 (Molecular Instruments PRF991), Stk26-B1 (Molecular Instruments PRC152) and Stk32a-B1 (Molecular Instruments PRF996).

## Quantification of fluorescence intensity

For quantification of mRNA detected by fISH (HCR v3.0) images were captured in three rectangle regions of interest (ROIs) of identical sizes were imaged with identical Z-plane depth from the lateral and medial regions of the utricle (*Figure 2C*) and processed as a max intensity projection using Zen software (Zeiss). Using ImageJ, the integrated density of each ROI was calculated for individual channels to quantify mRNA expression levels, and values were averaged across the three ROIs for a single biological replicate. Utricles from a minimum of three mice were analyzed for each genotype, and statistical significance was determined using student's *t*-test.

To quantify GPR156 in cochlear explants, we defined a 2x0.5 µm window in Fiji and positioned it on an medial or lateral outer hair cell junctional region where GPR156 signals were the most intense in the cell. The mean grey value was measured for each electroporated cell along with two adjcent un-electroporated control cells. GPR156 enrichment was calculated as a ratio of averaged values for

control cells. Statistical significance was determined using 1-way ANOVA (medial junction *Figure 7E*) or an unpaired Student's t-test (lateral junction *Figure 7F*).

## Preparation of AAV

HEK293 (ATCC, CRL-1573) cells were used for AAV preparation based upon an established protocol (*Challis et al., 2019*). Cells were grown in Gibco DMEM GlutaMAX (ThermoFisher Scientific 10569010) supplemented with 10% FBS. Cells were transiently transfected with three plasmids using Polyethylenimine (PEI, Polysciences Inc 23966–100). These plasmids were variants of pAAV-MCS; the transgene expression vector containing the chicken β-actin/CMV hybrid promoter CAG2 followed by our gene of interest, pHELPER; which encodes adenoviral proteins necessary for AAV genome replication and pAAV-ie which contains viral replication and capsid proteins optimized for inner ear transduction (*Tan et al., 2019*). HEK293 cells were seeded onto 150 mm culture dishes and grown to 80% confluency before carrying out transfection. Cells were transfected with 40 µg total DNA per 150 mm dish and the plasmid mixture was a ratio of 1:4:2 (pAAV-MCS:pAAVie:pHELPER). Culture media containing shed AAV was harvested 72 hr and 120 hr post transfection. The AAV particles were precipitated from the media by using Polyethylene Glycol (PEG, Sigma-Aldrich P2139). At 120 hr, cells were also harvested using a cell scraper and lysed with HL-SAN salt-active nuclease (ArticZymes 70910202) to release viral particles. Cell lysates and the PEG-precipitated virus were combined and AAV was purified on an Iodixanol (OptiPrep, Serumwerk Bernburg 1893) density gradient comprised of four layers (15%, 25%, 40%, and 60%). The combined lysate was loaded on top and then ultracentrifuged at 250,000x*g* for 3 hours at 18 C. After the centrifugation, the 40% layer containing AAV was collected, mixed with DPBS, and filtered through a 0.22 um syringe filter unit. Residual iodixanol was removed by washing with DPBS containing 0.001% pluronic F-68 and concentrating using 100 kDa MWCO centrifugal filter units (Millipore UFC910024). Purified virus was aliquoted and stored at –80 °C. Virus titer was determined by qPCR using SYBR Green and StepOnePlus (AppliedBiosystems) quantitative PCR machine as described (*Aurnhammer et al., 2012*). Variants of the pAAV plasmids used contain the CAG2 promoter followed by the coding sequence of gene(s) to be overexpressed. Each was sequence verified following construction.

## Utricle explant cultures

Utricles were dissected from the inner ears of P0 CD1 mouse pups in pre-chilled dissection solution (DPBS supplemented with 10% FBS, 0.1X N2 Supplement ThermoFisher 17502048) and 10 µg/ml Ciprofloxacin (SigmaAldrich 17850), transferred to 100 µl culture media (DMEM/F12 1:1 (ThermoFisher Cat 11330032) with 10% FBS, 0.1X N2 Supplement and 10 µg/ml Ciprofloxacin), and attached to a sterilized coverslip using Cell-Tak Cell and Tissue Adhesive (Corning 354240). Coverslips were placed into individual wells of a six-well plate and transduced with AAV premixed in culture media with AAV at (6X10$^7$ virions per µl) and incubated in a cell culture incubator at 37 °C for 4 hr. After the incubation, 2–3 ml fresh culture medium was added to each well and the explant was cultured for 3 days, with fresh media added after 48 hr. Following culture, explants were washed with PBS, then fixed for 30 min with 4% PFA/PBS on ice. After fixation and brief PBS wash, coverslips were superglued onto glass microscope slides with the explant facing up and processed for wholemount immunolabeling as described previously. Following labeling explants were coverslipped using Prolong Gold Antifade mounting media with the explant was sandwiched between the two glass coverslips and the bottom most coverslip superglued to the glass slide.

## Inner ear electroporation and cochlear culture

Inner ears from wild-type animals (C57BL/6J x FVB/NJ) were collected at E13.5 in HBSS +5 mM Hepes buffer (14065–056 and 15630–080, respectively, Gibco). Plasmid DNA was mixed with Fast Green FCF (final 0.05%; F7252, Sigma) and injected at 2 µg/µl into the cochlear duct using the Wiretroll plunger system (53507–426, Drummond). Next, the whole inner ear was electroporated (27 V, 27ms, 6 square pulses at 950ms intervals; BTX ECM 830), and the membranous labyrinth was dissected away from the condensed mesenchyme and embedded in Matrigel (8 µl drop of 50% Matrigel in DMEM; CB40234, Corning). The explants were cultured for 6 days in DMEM with 10% fetal bovine serum and 10 µg/ml ciprofloxacin (17850, Sigma), and then fixed in 10% TCA for 10 min on ice before being processed for immunolabeling.

## Measurements of stereociliary bundle orientation

To evaluate bundle orientation, utricles, saccules, and cristae were immunolabeled with antibodies against βII-SPECTRIN (1:1000; BD Biosciences 612562) and Oncomodulin/OCN (1:250; Santa Cruz Biotechnology SC-7466) as described above. Orientation was determined based on the polarized position of the fonticulus and measurements aided with a customized Python script (*Duncan et al., 2017*) that allowed orientations to be measured relative to a user-defined reference-line. The position of analysis fields for each sensory epithelia are defined in *Figure 4—figure supplement 2*. For the lateral utricle, the reference line was drawn along the utricle boundary while for medial striolar and MES regions the reference line was drawn along the boundary Oncomodulin labeling. Measurements for the saccule and cristae used reference-lines drawn along the boundary of the sensory epithelia. For circular histograms, measurements were pooled from three mutants and littermate controls and generated using Oriana circular graphing software (Kovach Computing Services).

## Quantitative RT-PCR

Inner ears were dissected from individual mice and coarsely cropped to separate the cochlea and semi-circular canals from the utricle and saccule, and this vestibular portion was transferred to individual Eppendorf tubes and flash frozen in liquid nitrogen. Following PCR genotyping, RNA was extracted from three knockout and three littermate control samples using the QIAGEN RNeasy Micro Kit (QIAGEN 74004) according to the manufacturers protocol. RNA concentration was determined by Nanodrop spectrophotometry and 2.5 µg of RNA was used to prepare cDNA using the Invitrogen SuperScript III Reverse Transcriptase (ThermoFisher Scientific 18080093). Quantitative RT-PCR was performed in a StepOnePlus Real-Time PCR System (Applied Biosystems) and PCR reactions contained 0.6 µM forward and reverse primer, 0.2 µL of cDNA and 2 X Power SYBR Green master mix. Following an incubation at 95 °C for 10 min, the reactions cycled between 95 °C denaturation for 15 s, and 58 °C extension for 1 min 40 times, and a melting curve was generated after the final cycle to verify the amplification specificity. Relative fold changes of gene expression were normalized by against the hair cell gene *Pou4f3* since hair cell number would not vary with the size of the cropped vestibule. For each primer set, three biological replicates and two technical replicates were used. PCR amplifications used the following primers: *Stk32a* primer C (5'- GACTTGACCATGGGAGCCAA –3'), *Stk32a* primer D (5'-TGGTGTCATTCTTCCGCACA –3') *Stk32a* primer E (5'-GGCCCTGGACTACCTACAGA-3'), *Stk32a* primer F (5'-TAAGGCTTGGTGCCAGCTAC-3'), *GAPDH* for (5'-TGGAGCCAAAAGGGTCA-3'), *GAPDH* rev (5'-CTTCTGGGTGGCAGTGA-3'), *Pou4f3* for (5'-CCAGACTCCCGAAGATGATGG-3'), *Pou4f3* rev (5'-GCCAGCAGGCTCTCATCAAA-3').

## Statistical analysis

All data is graphed as mean +/- SEM or SD as indicated in figure legends and statistical analysis was conducted using GraphPad Prism 9. The student's *t*-test was used to analyze the comparison and a p-*value* <0.05 was considered to be statistically significant. For measures of GPR156 medial and lateral localization, ne-way ANOVA and unpaired t-test were respectively used to analyze the comparison and p-*values* <0.05 was considered to be significant. Gpr156 data is graphed as 25–75% boxplots where exterior lines show minimum and maximum values, the middle line the median and +the mean. For analysis of RNAseq datasets, Benjamini-Hirshberg false discover rate was used to normalize data across all transcripts and produce adjusted p-values for significantly downregulated or upregulated genes based on log2-fold changes compared to WT baseline transcript levels. Unless otherwise stated, fluorescent imaging was completed on a minimum of three biological replicates with higher quality representative images selected for publication.

## Acknowledgements

This work was supported by the National Institutes of Health (R01DC013066 to MRD, R01DC015242 and R01DC018304 to BT, and Eunice Kennedy Shriver National Institute of Child Health & Human Development of the NIH Award #T32HD007491 to EMR). We thank Doris Wu (NIDCD) for sharing reagents.

## Additional information

### Funding

| Funder | Grant reference number | Author |
|---|---|---|
| National Institutes of Health | R01DC013066 | Michael R Deans |
| National Institutes of Health | R01DC015242 | Basile Tarchini |
| National Institutes of Health | R01DC018304 | Basile Tarchini |
| National Institutes of Health | T32HD007491 | Evan M Ratzan |

The funders had no role in study design, data collection and interpretation, or the decision to submit the work for publication.

### Author contributions

Shihai Jia, Formal analysis, Validation, Investigation, Visualization, Methodology, Writing – original draft, Data curation; Evan M Ratzan, Conceptualization, Formal analysis, Investigation, Methodology, Writing – review and editing; Ellison J Goodrich, Raisa Abrar, Luke Heiland, Investigation; Basile Tarchini, Resources, Formal analysis, Validation, Investigation, Visualization, Methodology, Writing – review and editing, Funding acquisition; Michael R Deans, Conceptualization, Formal analysis, Supervision, Funding acquisition, Validation, Investigation, Visualization, Writing – original draft, Project administration, Writing – review and editing

### Author ORCIDs

Shihai Jia ⓘ http://orcid.org/0000-0003-1144-4291
Basile Tarchini ⓘ http://orcid.org/0000-0003-2708-6273
Michael R Deans ⓘ http://orcid.org/0000-0001-6319-7945

### Ethics

All animal work was reviewed for compliance and approved by the Animal Care and Use Committee of The University of Utah (IACUC protocol #00001498) and the Animal Care and Use Committees of The Jackson Laboratory (Animal Use Summary AUS no. 14012).

### Decision letter and Author response

Decision letter https://doi.org/10.7554/eLife.84910.sa1
Author response https://doi.org/10.7554/eLife.84910.sa2

---

## Additional files

### Supplementary files

• Supplementary file 1. Down-regulated genes in the *Emx2* Cre/Cre utricle. Summary of genes that are down-regulated in the absence of *Emx2*. Sequencing results were analyzed using Bioconductor DESeq as well as post hoc-analysis using Swish, and Benjamini-Hirshberg false discover rate was used to normalize data across all transcripts and produce adjusted p-values based on log2-fold changes between *Emx2* Cre/Cre and WT controls. Only genes with adjusted *P*-values <0.05 are listed. Additional data for genes highlighted with blue shading are presented in *Figure 1—figure supplement 1*.

• Supplementary file 2. Up-regulated genes in the *Emx2* Cre/Cre utricle. Summary of genes that are up-regulated in the absence of *Emx2*. Sequencing results were analyzed using Bioconductor DESeq as well as post hoc-analysis using Swish, and Benjamini-Hirshberg false discover rate was used to normalize data across all transcripts and produce adjusted p-values based on log2-fold changes between *Emx2* Cre/Cre and WT controls. Only genes with adjusted *P*-values <0.05 are listed. Red shading highlights *Stk32a*.

• MDAR checklist

## Data availability

Sequencing data have been deposited in GEO under accession code GSE218746.

The following dataset was generated:

| Author(s) | Year | Dataset title | Dataset URL | Database and Identifier |
|---|---|---|---|---|
| Ratzan EM, Paul L, Stubbens C, Dalley B | 2022 | Emx2 KO RNAseq | https://www.ncbi.nlm.nih.gov/geo/query/acc.cgi?acc=GSE218746 | NCBI Gene Expression Omnibus, GSE218746 |

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
