## [Editor Report]

This important study provides a significant advance in the understanding of the molecular mechanisms that establish planar cell polarity in hair cells of the mammalian inner ear. The conclusions, which are supported by compelling evidence, will be of interest to those studying the development and function of the vestibular system, and planar cell polarity.

---

## [Decision Letter]

**Decision letter after peer review:**

Thank you for submitting your article "The dark kinase STK32A regulates hair cell planar polarity opposite of EMX2 in the developing mouse inner ear" for consideration by *eLife*. Your article has been reviewed by 3 peer reviewers, and the evaluation has been overseen by a Reviewing Editor and Marianne Bronner as the Senior Editor. The following individuals involved in the review of your submission have agreed to reveal their identity: Hernán López-Schier (Reviewer #1); Brian McDermott (Reviewer #3).

Essential revisions:

1) Demonstrate whether or not Stk32a is a direct transcriptional target of Emx2, or tone down this conclusion.

2) Provide further analysis of the nature of the Stk32a mutation, including whether or not any protein is produced.

3) Test whether the kinase activity of Stk32a is required for its function.

4) Either test whether Stk32a directly interacts with Gpr156, or discuss whether Stk32a regulates Gpr156 via direct or indirect mechanisms.

*Reviewer #1 (Recommendations for the authors):*

Figures are truncated in my version of the manuscript.

I do not understand how the authors can claim that the Stk32a-/- mice have massively disrupted vestibular hair cells, but have no discernible behavioural differences from controls.

Please revise the idea that Stk32a is a direct target of Emx2.

*Reviewer #2 (Recommendations for the authors):*

Specific comments:

1. Since mutant mRNA perdures in the Stk32a mutant utricle (Figure S3), it would be important to more clearly define the nature of the mutation and determine whether any Stk32a protein is produced in the mutant utricle by Western blot. Multiple WB-validated Stk32a C-terminal antibodies are commercially available and can be tested (e.g. Σ SAB1405383). In addition, it should be clarified whether Stk32a heterozygous maculae had any PCP defects, to rule out potential dominant-negative effects.

2. How might Emx2 repress Stk32a transcription? Is this regulation direct or indirect? Was there a precedent for Emx2 as a repressor? This should at least be discussed if not experimentally addressed. Of note, ectopic expression of Emx2 in the MES did not appear to repress Stk32a expression as effectively as in the lateral region (Figure 2F), suggesting that additional factors may be involved to fully repress Stk32a.

3. Several outstanding questions remain regarding the mechanism by which Stk32a regulates GPR156 surface expression, some of which are well within the scope of the current study. For example, is the kinase activity of Stk32a required for its function? It should be entirely feasible to test whether a Stk32a kinase-dead construct can reduce GPR156 levels in cochlear explants, as shown in Figure 7 (and ideally also in utricle explants as shown in Figure 3). The kinase-dead construct would also help distinguish whether Stk32a overexpression reorients PCP via dominant-active or dominant-negative effects.

4. Related to the above, the data and model really beg the question of whether Stk32a directly interacts with Gpr156. It should be feasible to test this in heterologous cells transfected with Gpr156 and GFP-tagged Stk32a wt or kinase-dead constructs using co-IP assays.

---

## [Author Response]

Essential revisions:1) Demonstrate whether or not Stk32a is a direct transcriptional target of Emx2, or tone down this conclusion.

We appreciate this concern and recognize that our experiments do not distinguish between a direct or indirect relationship between EMX2 and *Stk32a* transcription. As recommended, we have made text edits throughout the manuscript that are meant to specifically tone done the conclusion that Stk32a is a direct target of Emx2. Notable changes include:

2) Provide further analysis of the nature of the Stk32a mutation, including whether or not any protein is produced.

We appreciate this recommendation as well as the underlying concern that the KO phenotype reported here may not be null, and that possible expression of mutant protein may provide residual function or even a dominant phenotype. We have tested multiple commercial antibodies (Σ, ProteinTech, RayBiotech), and have contracted the production of our own custom antibodies yet remain unable to identify a reagent with suitable specificity or affinity to detect STK32A by Western Blot or Immunofluorescence assays.

In the absence of a suitable reagent we have provided the following additional analyses of the *Stk32a* mutation that support our interpretation that the *Stk32a* KO is a null, and that any mutant protein, if produced, is not functional. We feel that these analyses should be sufficient because in the event that we had detected mutant protein via Western blot, these are the experiments we would have conducted to determine whether or not that mutant protein had an effect.

We have prepared a new figure supplement demonstrating that no phenotype emerges in Stk32a heterozygotes and that the organization of vestibular hair cells in the utricle is not changed between wild type and *Stk32a*
^+/-^ littermates. Thus, targeted deletion of *Stk32a* exon2 does not result in a dominant phenotype. Please see Figure 4 —figure supplement 3 in the resubmitted manuscript. Moreover, this addition allows us to clarify an experimental detail that was not clearly stated during the first submission; that the ‘Control’ mice used for the KO analysis were all heterozygous for the *Stk32a* exon2 deletion.We have added additional experimentation demonstrating that any mutant protein that might be produced from the *Stk32a* mutant allele is not functional. In brief, we demonstrate that a predicted myristoylation site at the second amino acid (Glycine) of STK32A is necessary for GFP-tagged protein localization to the apical surface of hair cells. Furthermore, deletion of this Glycine prevents stereociliary bundle reorientation following AAV-mediated expression of GFP-tagged *Stk32a*
^Δ2G^ mutant protein in Emx2-positive hair cells demonstrating that myristoylation is essential for this STK32A function.

Since this amino acid is encoded by exon2 and is deleted in the *Stk32a* KO we anticipate that any mutant proteins translated from alternative start codons will also be non-functional.

3) Test whether the kinase activity of Stk32a is required for its function.

We have improved the resubmitted manuscript as recommended by the reviewers by conducting additional experiments using a mutant *Stk32a* construct that is predicted to disrupt kinase function. This mutation replaces an essential Lysine at amino acid position 52 with Arginine (*Stk32a ^K52R^*). Based upon sequence similarities to related kinases this Lysine is necessary for STK32A to bind ATP prior to substrate phosphorylation (Sorrell et al. 2020 and UniProt Consortium 2021) and substitution with Arginine disrupts kinase activity. This mutant has been used to demonstrate that the kinase activity of STK32A is required for function in the following experiments:

We have used plasmids expressing *Stk32a ^K52R^* to electroporate hair cells in the developing cochlea. These experiments have been added to Figure 7. In brief, kinase-dead Stk32a ^K52R^ does not impact GPR156 delivery or maintenance at the apical cell surface, indicating the kinase activity is necessary for this aspect of STK32A function.

Based upon these experiments we conclude that the kinase activity of Stk32a is required for its function and appreciate the recommendation to incorporate these additional experiments.

4) Either test whether Stk32a directly interacts with Gpr156, or discuss whether Stk32a regulates Gpr156 via direct or indirect mechanisms.

The potential for physical interactions between STK32A and GPR156 has been evaluated in heterologous cell lines by co-transfecting GST and GFP tagged constructs. This approach has not revealed a direct physical interaction, however negative results using these approaches have little meaning since we cannot exclude the possibility that intermediaries or cofactors present in hair cells, but not in cultured cells, facilitate the proposed interactions. As such we have decided not included those studies in this manuscript. Instead, as recommended by the reviewers, we have discussed the possibility that STK32A regulation of GPR156 may be indirect.

Reviewer #1 (Recommendations for the authors):Figures are truncated in my version of the manuscript.

This is unfortunate and, if given the opportunity, we will ensure that each figure is appropriately formatted prior to publication.

I do not understand how the authors can claim that the Stk32a-/- mice have massively disrupted vestibular hair cells, but have no discernible behavioural differences from controls.

We were also surprised that changes in planar polarity that are so readily detected based upon vestibular hair cell anatomy did not impact the mouse’s behavior in its home cage. It is possible that deficits will be revealed by more advanced behavioral testing but that is beyond the scope of this developmental study.

Please revise the idea that Stk32a is a direct target of Emx2.

We have made changes to revise this idea and discuss the possibility of an intermediary between EMX2 and *Stk32a* repression. These changes were described in detail in response to Essential Revision #1.

Reviewer #2 (Recommendations for the authors):Specific comments:1. Since mutant mRNA perdures in the Stk32a mutant utricle (Figure S3), it would be important to more clearly define the nature of the mutation and determine whether any Stk32a protein is produced in the mutant utricle by Western blot. Multiple WB-validated Stk32a C-terminal antibodies are commercially available and can be tested (e.g. Σ SAB1405383). In addition, it should be clarified whether Stk32a heterozygous maculae had any PCP defects, to rule out potential dominant-negative effects.

Despite the availability of several commercially available antibodies we have not identified one that can detect STK32A protein from mouse tissues by either Western blot or immunofluorescent labeling.

Nonetheless, we have taken two steps to address these important concerns as presented in greater detail in Essential Revision #2.

We demonstrate that an N-myristoyl Glycine at amino acid position 2 is necessary for STK32A localization and function. Since this amino acid is encoded by exon2 which is deleted in the *Stk32a* KO mouse we do not expect that any mutant protein, if produced, to be functional in the KO.We demonstrate by comparing wild type and Stk32a heterozygous hair cells that there are no PCP defects and thus no dominant-negative effects of any mutant protein that may be produced.

2. How might Emx2 repress Stk32a transcription? Is this regulation direct or indirect? Was there a precedent for Emx2 as a repressor? This should at least be discussed if not experimentally addressed. Of note, ectopic expression of Emx2 in the MES did not appear to repress Stk32a expression as effectively as in the lateral region (Figure 2F), suggesting that additional factors may be involved to fully repress Stk32a.

There is a well-established precedent for EMX2 acting as a repressor during the course of telencephalic development and cortical arealization. However, much of this work is based upon changes in gene following *Emx2* mutation or overexpression (similar to our own study). Notable exceptions occur in the regulation of *Sox2* expression where EMX2 binds POU sites in the *Sox2* enhancer which has been proposed to repress gene expression in a dose dependent manner (Mariani et al. 2012), and in the regulation of *Gsx2* expression where EMX2 functions as a repressor in the presence of cofactors (Desmaris et al. 2018). We have added a paragraph to the Discussion section that presents these examples, while clearly stating that our own study does not distinguish between direct or indirect modes of regulation.

3. Several outstanding questions remain regarding the mechanism by which Stk32a regulates GPR156 surface expression, some of which are well within the scope of the current study. For example, is the kinase activity of Stk32a required for its function? It should be entirely feasible to test whether a Stk32a kinase-dead construct can reduce GPR156 levels in cochlear explants, as shown in Figure 7 (and ideally also in utricle explants as shown in Figure 3). The kinase-dead construct would also help distinguish whether Stk32a overexpression reorients PCP via dominant-active or dominant-negative effects.

We have conducted additional experiments using a kinase-dead STK32A construct that test its function in both cochlear and utricle explants. These changes were described in detail in response to Essential Revision #3.

4. Related to the above, the data and model really beg the question of whether Stk32a directly interacts with Gpr156. It should be feasible to test this in heterologous cells transfected with Gpr156 and GFP-tagged Stk32a wt or kinase-dead constructs using co-IP assays.

We have conducted experiments to test the potential for STK32A and GPR156 to physically interact or for STK32A to phosphorylate GPR156 following co-transfection of heterologous cell lines. To date these efforts have only yielded negative results. Unfortunately, an unavoidable flaw in this approach is that it is not known whether or not the cell lines we employed express cofactors or intermediaries needed for these interactions to occur (e.g. additional kinases that activate or are activated by STK32A) which makes interpretation of a negative result impossible. For these reasons those results have not been included in the current manuscript.